



# Sensitivity of subglacial drainage to water supply distribution at the Kongsfjord basin, Svalbard

Chloé Scholzen[1], Thomas V. Schuler[1], Adrien Gilbert[2]

[1]Department of Geosciences, University of Oslo, Norway
[2]Université Grenoble-Alpes, CNRS, IGE, Grenoble, France

*Correspondence to*: Chloé Scholzen (chloe.scholzen@geo.uio.no)

**Abstract.** By regulating the amount, the timing and the location of meltwater supply to the glacier bed, supraglacial hydrology potentially exerts a major control on the evolution of the subglacial drainage system, which in turn modulates sliding. Yet the configuration of the supraglacial hydrological system has received only little attention in numerical models
of subglacial hydrology so far. Here we apply the two dimensional subglacial hydrology model GlaDS to a Svalbard glacier basin with the aim of investigating how the spatial distribution of meltwater recharge affects the characteristics of the basal drainage system. We design four experiments with various degrees of complexity in the way that meltwater is delivered to the subglacial drainage model. Our results show significant differences between experiments in the early-summer transition from distributed to channelized drainage, with discrete recharge at moulins favouring channelization and driving earlier rise
in basal water pressure. Otherwise, we find that water input configuration only poorly influences subglacial hydrology, which is controlled primarily by subglacial topography. All experiments fail to develop channels of sufficient efficiency to substantially reduce summertime water pressures, which we impute to small surface gradients and short melt seasons. The findings of our study may be extended to most Svalbard tidewater glaciers with similar topography and low meltwater recharge. The absence of efficient channelization implies that the dynamics of tidewater glaciers in the Svalbard archipelago
may be sensitive to future long-term trends in meltwater supply.

## 1 Introduction

Land-based ice masses, as they undergo rapid change due to climate warming, are one of the largest potential contributors to global sea level rise (AMAP, 2017; Pörtner et al., 2019, Wouters et al., 2019). Regional glacier change also affects surface albedo, coastal ecology, as well as hydrological management in terms of flood hazards, hydropower and freshwater supply
(Vincent et al., 2011; Fountain et al., 2012; Carey et al., 2017; Milner et al., 2017). As such, predicting and adapting to future alterations in the glacial landscape relies on better understanding glacier response to changing climate. Field observations and modelling experiments suggest that, besides increasing mass loss, higher meltwater production may also alter ice dynamics (Hewitt, 2013; Gagliardini and Werder, 2018; Davison et al., 2019). Seasonal ice flow accelerations following periods of enhanced surface melt have been reported for glaciers worldwide (e.g. Iken et al., 1983; Mair et al., 2001; Zwally




et al., 2002; Copland et al., 2003), including polythermal glaciers in Svalbard (Dunse et al. 2012, 2015; Schellenberger et al., 2015). On shorter timescales (i.e. hourly or daily), episodic speedups have been found concurrent with intense melting, heavy rainfall, and supraglacial lake drainage (e.g. Joughin et al., 2013; Horgan et al., 2015).

Both theoretical and observational studies have long established that meltwater reaching the glacier bed modulates basal water pressure, which in turn affects ice motion through basal sliding (e.g. Lliboutry, 1968; Bindschadler, 1983; Iken and

Bindschadler, 1986; Sugiyama and Gudmundsson, 2004). The relationship between water input and sliding is nevertheless complicated by the morphology of the subglacial hydrological system (Bartholomew et al., 2010; Sole et al., 2011; Sundal et al., 2011; Cowton et al., 2013). High basal water pressures and acceleration are typically related to subglacial flow in distributed linked cavities (inefficient drainage) (Kamb, 1987; Iken and Truffe, 1997), whereas lower water pressures and ice flow deceleration are imputed to channelization (efficient drainage) (Röthlisberger, 1972; Nienow et al., 1998; Schoof,

40   2010).

Hydrological processes taking place inside and below glaciers are of substantial complexity and remain poorly understood due to the sparsity of direct observations. To address this lack of knowledge, numerous modelling approaches have been developed in the past decades, with much of the recent effort directed at the coupling between glacier dynamics and basal water pressure (e.g. Schoof, 2010; Hewitt, 2013; de Fleurian et al., 2014; Hoffman and Price, 2014; Bueler and

Van Pelt, 2015). However, so far only a handful of these studies have focused on the role of supraglacial hydrology in the evolution of the subglacial drainage system (Clason et al., 2015; Wyatt et al., 2015; Banwell et al., 2016; Koziol et al., 2017; Decaux et al., 2019). As pointed out by Flowers (2015), most conceptual models of subglacial hydrology implicitly assume that recharge of surface water to the glacier bed is homogeneous (e.g. Flowers and Clarke, 2002; Hewitt, 2011; Pimentel and Flowers, 2011). In nature, water supply to the subglacial environment is rarely uniform and usually collects meltwater

produced in an upstream area of the glacier surface. In the accumulation zone, recharge to the bed is delayed as the surface-generated water percolates through the snow and firn pack, where some fraction may be temporarily retained as slush, or may refreeze if subsurface temperatures are below melting point (Pfeffer et al., 1991; Van Pelt et al., 2016). The water that escapes the firn is conveyed by englacial conduits to the subglacial system (Gulley et al., 2009a, 2009b; Benn and Evans, 2010). In the bare-ice zone, surface meltwater collects in supraglacial streams and is routed downglacier over hundreds of

meters to several kilometres. Within one melt season, surface meltwater either enters a crevasse, a moulin or a supraglacial lake, or leaves the glacier by running off the glacier edge. Both crevasses and moulins are important surface-to-bed hydrological connections as they can be found over the entire glacier. Moulins are near-vertical conduits that result from water pouring into a crevasse and melt-enlarging it through frictional heating. As they are typically fed by supraglacial streams, moulins capture runoff from upstream watersheds. By transferring significant volumes of water to the glacier bed at

discrete points, moulins can initiate channels in subglacial regions that may not be predicted by the hydraulic potential minimum (Gulley et al., 2012). Moulins persist over multiple years if they receive frequent water delivery from the supraglacial system, and therefore may have a long-lived impact on the subglacial hydrology (Catania and Neumann, 2010). Consequently, explicit representation of water recharge to the base of glaciers may be crucial to accurately simulate how the





subglacial hydrological system, and hence ice dynamics, respond to increased meltwater production and variability in a
warming climate.

Here we apply a two dimensional distributed model of subglacial hydrology (GlaDS) (Werder et al., 2013) to a Svalbard polythermal glacier basin. The model, which couples distributed and channelized drainage, is forced with distributed supply rates from a calibrated surface-energy-balance-snow model (Van Pelt et al., 2019). We define different water input configurations that include spatially continuous and/or discrete subglacial water recharge with aims to: (a) investigate how
approximations on water supply to the bed affect the development of the modelled subglacial hydrological system, and (b) describe the basal drainage system at the Kongsfjord basin, with special focus on basal water pressure distribution, drainage efficiency, and its implications for glacier sliding.

## 2 Study area

### 2.1 Glaciers and topography

The Kongsfjord basin, located in northwestern Svalbard, covers a total land area of approximately 1,430 km$^2$, of which about 75 % is glacierised. The basin surrounds Kongsfjorden, which stretches over ~20 km towards the Arctic Ocean to the west. This study centers on three adjacent glacier systems that terminate in the inner part of Kongsfjorden (Fig. 1(a)): Isachsenfonna/Kongsbreen, Holtedahlfonna-Infantfonna/Kronebreen, and Kongsvegen.

At the northeastern fjord head, the ice field Isachsenfonna drains through the outlet glacier Kongsbreen (together 378
km$^2$), which splits due to a bedrock ridge into two branches, Kongsbreen North and Kongsbreen South. To the east, the terminal glacier Kronebreen is fed by two contributory catchments: the larger ice field Holtedahlfonna and the smaller Infantfonna (together 373 km$^2$). Roughly 5 km from its front, Kronebreen is confluent with Kongsvegen (108 km$^2$), and the two glacier tongues share a common terminus which is currently dominated by Kronebreen (Sund et al., 2011).

Surface elevation (Fig. 1(a)), basal topography (Fig. 1(b)), and ice thickness of the Kongsfjord basin were recently
mapped by airborne and ground-based ice-penetrating radar (Lindbäck et al., 2018). Together with their accumulation areas, the Kongsfjord glaciers span an altitudinal range from sea level to 1,400 m a.s.l. in the northern parts of their catchments (Fig. 1(a)). Below the equilibrium line altitude, which for Holtedahlfonna/Kronebreen was estimated around 610 m a.s.l. for the period 1961–2012 (Van Pelt and Kohler, 2015), Kronebreen is heavily crevassed and its slope is disrupted by several ice falls. In the ablation area, the subglacial topography is characterised by extended troughs as deep as −180 m a.s.l. under
Kongsbreen North, −130 m a.s.l. under Kronebreen, and −70 m a.s.l. under Kongsvegen (Fig. 1(b)). Both Kronebreen and Kongsvegen are fully grounded below sea level over their lowermost 10 km and 9 km, respectively, whereas Kongsbreen South is mostly resting on bedrock above the water line. A few nunataks peak through the lower parts of the glaciers at approximately 500–600 m a.s.l.. Further inland, the subglacial topography smoothens into wide flat valleys around 200 m a.s.l., and ice thickens with a maximum of 740 m in the middle of Isachsenfonna.



## 2.2 Mass balance and dynamics

Like most tidewater glaciers in Svalbard, the Kongsfjord glaciers have a negative mass balance dominated by calving, a finding that both in situ measurements and modelling experiments have corroborated over the last decades (e.g. Lefauconnier et al., 1994b; Nuth et al., 2012). More recently, Van Pelt and Kohler (2015) used a coupled surface-energy-balance-snow model to include superimposed ice and internal accumulation in their surface mass balance calculations. For Kronebreen and Kongsvegen, they found that surface mass balance is controlled by mass gain through precipitation (0.87 m w.e. $a^{-1}$) and mass loss through runoff (0.79 m w.e. $a^{-1}$), and that refreezing of water in the firn layer (0.30 m w.e. $a^{-1}$) provides a strong buffer for runoff. Their results also reveal a 31 % increase in runoff since 2000, coinciding with higher surface melt rates and lower refreezing rates in response to recent warming and firn line retreat.

Kronebreen is one of the fastest-flowing glaciers in Svalbard (Liestøl, 1988; Sund et al., 2011). Near the terminus, mean annual surface velocities are persistently high (1.6–2.15 to m $day^{-1}$) (Lefauconnier, 1987; Lefauconnier et al., 1994a; Kääb et al., 2005), and summer velocities peak up to 3.2 m $day^{-1}$ (Schellenberger et al., 2015). Such high velocities can only be achieved through basal sliding, which indicates that the glacier base is temperate. Kongsbreen has a more complex velocity pattern, due to its split into two separate terminal branches: the northern, marine-terminating branch of the glacier is fast-flowing, with a maximal speed of 2.7 m $day^{-1}$ measured in 2012 (Schellenberger et al., 2015), whereas the southern, partially land-terminating branch is slow-moving. Both Kronebreen and Kongsbreen North exhibit clear seasonal flow variability characterised by relatively stable and low velocities in autumn and winter, and an acceleration synchronous with the melt season. Schellenberger et al. (2015) observed that, at both glaciers, a large part of the ice flow variability followed fluctuations in surface meltwater production and rainfall, a correlation that they attributed to variations in basal lubrication. Evidence from later studies also strongly suggest that multiannual changes in ice flow at Kronebreen are controlled by changes in the subglacial hydrology (How et al., 2017; Vallot et al., 2017). Interannual variability in ice velocity as well as upglacier propagation of seasonal speedup were both found to be higher at Kronebreen than at Kongsbreen North (Schellenberger et al., 2015), indicating different basal conditions at the two glaciers. Kongsvegen, a surge-type glacier that is currently in its quiescent phase, contrasts with the other Kongsfjord glaciers in its dynamics. Since its last surge in 1948 (Melvold and Hagen, 1998), the glacier has experienced very little motion (2–8 m $a^{-1}$), negligible calving and significant surface steepening (Hagen et al., 1999; 2005; Nuth et al., 2012).

## 2.3 Subglacial hydrology

Glaciers in Kongsfjord are polythermal, usually with basal temperatures at or close to the pressure melting point, meaning that water drains at their base all year through (Björnsson et al., 1996; Sevestre et al., 2015; Nuth et al., 2019). Glacial runoff is periodically released into the fjord, where it produces sediment-rich freshwater plumes that are of great importance for the glacio-marine dynamics and the ecosystem of Kongsfjorden (Svendsen et al., 2002; Everett et al., 2018).





In recent years, several locations of subglacial discharge have been identified at the Kronebreen-Kongsvegen terminal complex: one main outlet in the northern half of the glacier tongue, and several secondary outlets near the Kronebreen-Kongsvegen junction (Trusel et al., 2010; Kehrl et al., 2011; Everett et al., 2018). Contrasts in plume timing and duration have been suggested to reflect local differences in subglacial drainage efficiency across glacier termini (Schild et al., 2016;

Slater et al., 2017). How et al. (2017) monitored plumes at Kronebreen and combined the data with observations of borehole water pressure, supraglacial lake drainage, surface velocities, as well as modelling of meltwater routing, to infer the morphology of the subglacial hydrological system during summer 2014. Their results indicated that meltwater was largely drained to the northern part of the terminus through a channelized system connected to the upper catchment area (Holtedahlfonna), hence the broader and more persistent plume on the northern side. By contrast, the intermittent activity of

the southern plume during the 2014 melt season was attributed to distributed drainage of a smaller catchment area.

Spatiotemporal variations in basal properties at Kronebreen were also investigated by Vallot et al. (2017) through inverting surface velocities for 2013–2015 to determine a basal friction coefficient. Their modelling results supported conclusions drawn by the earlier studies, i.e. that subglacial hydrology organisation is as much a key factor to basal sliding as are local characteristics such as bed topography and summer melt.

While the relationship between meltwater supply, subglacial hydrology and ice velocity is well documented at Kronebreen (Schellenberger et al., 2015; How et al., 2017), similar knowledge is lacking at the neighbouring outlet glaciers (Kongsvegen, Kongsbreen North and Kongsbreen South), and the subglacial hydrology at the upper ice fields (Isachsenfonna, Holtedahlfonna and Infantfonna) remains yet to be examined.

## 3 Methods and datasets

### 3.1 Mapping of supraglacial hydrology

Mapping of Kongsfjord's supraglacial drainage system is achieved using TopoToolbox, a collection of scripts to analyse flow pattern in digital elevation models (DEMs) (Schwanghart and Scherler 2014). TopoToolbox first seeks to determine surficial water movement within the DEM. Flow direction calculation is based on a multiple flow direction algorithm that examines elevation gradients between grid cells. The multiple flow direction algorithm allows for both convergence and

divergence of flow, thus producing more realistic flow pathways than the single flow direction algorithm. Next, information on flow direction is used to derive: (1) flow accumulation, i.e. the number of cells draining in each grid cell, (2) the drainage network, i.e. the estimated flow paths in the DEM, and (3) drainage basins, i.e. the upslope areas contributing to each grid cell in the drainage network. We generate a map of the supraglacial stream network, shown in Fig. 2(c), and the watersheds associated to each moulin (Fig. 2(c,d)). The selection of the moulin locations is described in Sect. 3.3.2.



## 3.2 Modelling of subglacial hydrology

To simulate the subglacial hydrology of the Kongsfjord basin, we employ the Glacier Drainage System model (GlaDS) (Werder et al., 2013), a distributed model developed to describe subglacial water drainage and that participated in the Subglacial Hydrology Model Intercomparison Project (SHMIP) (de Fleurian et al., 2018). GlaDS has been implemented in Elmer/Ice, an open source, finite-element model (Gagliardini et al., 2013), which in this study serves only as platform for the subglacial hydrology model. The implementation of GlaDS in Elmer/Ice is detailed in Gagliardini and Werder (2018).

The subglacial hydrology model GlaDS accounts for both inefficient distributed drainage and efficient channelized drainage to compute the evolution of the hydraulic potential at the glacier base. The distributed network of linked cavities is featured in the form of a continuous water sheet across the model domain, which is fed by a source term at each node on the mesh. We define the source term from the sum of a space and time varying surface water input, as described in Sect. 3.4.2, and a uniform, steady basal melting. Discrete channels that form on the edges of the mesh elements constitute the efficient drainage system. Between each pair of nodes, channel segments develop from water exchange with the adjacent sheet as well as from water input at moulins, which act as local sources. Moulins are shaped as vertical cylinders at some of the mesh nodes (see Sect. 3.3.2), and deliver water directly from the glacier surface to the basal channel network. Depending on their cross-sectional area and on the subglacial water pressure, moulins can store a certain volume of water, as revealed by tracer-experiments (Werder and Schuler, 2010). For a complete description of the equations solved by the GlaDS model, we redirect readers to Werder et al. (2013).

The problem is solved on a mesh constructed by discretising the domain using irregular triangulation with a mean edge length of 250 m, which yields 11,824 nodes in total. The mesh resolution is a compromise between numerical computation time and spatial precision of the model outputs. To ensure computational stability, the simulation time step does not exceed 1 day, and adaptive time stepping down to ~1.5 minutes is applied. Additional gain of computation time is achieved through partitioning and parallel simulation of each model run.

As boundary conditions, channels are not allowed to develop along the outer limits of the model domain, and the hydraulic potential is set to zero at the glacier fronts because we consider that all three termini (Kongsbreen North, Kongsbreen South, and Kronebreen-Kongsvegen) are freely connected to the fjord, which imposes the water pressure. The values for the model parameters are set out in Table 1. The grey highlighted parameters are those to which we assigned values different from the literature. We adapted values for conductivity parameters as well as for sheet width below channel to facilitate channelization and to maximise the influence of meltwater input on the subglacial drainage efficiency. In addition, we set a uniform basal sliding speed of ~315 m a$^{-1}$ for the entire domain, which corresponds roughly to the glacier-averaged surface speed measured by satellite imagery in the Kongsfjord basin (Schellenberger et al., 2015).



### 3.3 Water supply to the subglacial drainage system

In order to investigate the sensitivity of our subglacial hydrology model to water recharge distribution, we design four model experiments with various degrees of complexity in the way that water is delivered to the glacier beds. The control experiment is the simplest scenario, in which basal water supply is spatially continuous and identical to surface meltwater production. The next two experiments assume discrete water recharge to the basal drainage system through moulins, with the latter including only water that is drained in moulins catchments. Finally, the fourth and most realistic scenario builds upon local knowledge of glacial hydrology at Kongsfjord. This experiment accounts for immediate surface-to-base water transfer via crevasses in the ablation zone, as well as for catchment-wide water collection into moulins in the accumulation zone. Water input is taken from a raster of modelled recharge values, as described in Sect. 3.4.2 below.

#### 3.3.1 Experiment 1

In the first experiment, the gridded water input is interpolated to every mesh node of the GlaDS model and is directly injected into the distributed sheet layer, regardless of the supraglacial hydrology (Fig. 2(a)). This approximation is most commonly made in subglacial hydrology models, based on the assumption that glaciers are perfectly 'permeable' in that they allow meltwater to travel straight from their surface to their beds. We use this experiment as a reference against which we compare the other experiments with the aim of assessing the effect of adding complexity to water supply distribution.

#### 3.3.2 Experiment 2

In this scenario, water input is restricted to only a few moulins that act as individual sinkholes at the glacier surface. We evenly redistribute the total recharge between the moulins, so that at each time step every moulin receives the same amount of water. We consider 13 moulins in total, five of which have been previously identified as active supraglacial lakes that drained to the glacier base at least once between 2014 and 2017. In addition, based on high-resolution aerial images derived from TopoSvalbard ([https://toposvalbard.npolar.no/](https://toposvalbard.npolar.no/), Norwegian Polar Institute), we manually detect at least 20 more moulins, of which we keep only 10 that are in vicinity to the supraglacial stream network inferred by TopoToolbox (Fig. 2(c)). Each moulin is then defined at the closest mesh node of the model domain. All 13 moulins are located between approximately 3–35 km from the glacier termini, and at surface elevations between 400 and 700 m a.s.l. (Fig. 2(b)). Both the location and the cross-sectional area (see Table 1) of the moulins are assumed constant over the study period.

#### 3.3.3 Experiment 3

Here we impose discrete water input at the same 13 moulins as in Experiment 2, with the difference that water recharge is calculated from the catchment area that is associated to each moulin (Fig. 2(c)). Using the TopoToolbox program (described in Sect. 3.1 above), we delineate the supraglacial watersheds and retrieve the meltwater volume in each of them. The moulins catchments are assumed to remain stationary over the study period. The total water amount involved here is slightly



higher than in Experiments 1–2 because this scenario accounts for meltwater in watersheds that extend beyond the glacier boundaries.

### 3.3.4 Experiment 4

Finally, we develop an experiment in which we attempt to describe as realistically as possible the spatial distribution of water supply to the glacier beds. Here, we combine continuous and discrete recharge to the subglacial hydrological system.

In crevassed areas of the Kongsfjord basin, we apply direct surface-to-bed transfer of meltwater into the sheet layer, in the same fashion as for Experiment 1. The crevassed zones are manually mapped on 1 m resolution satellite images from Pleiades (© CNES (2014), and Airbus DS (2014), all rights reserved. Commercial uses forbidden). It should be noted that, whilst some minor cracks are also detected in Holtedahlfonna and Isachsenfonna, the extent of these crevassed areas is probably underestimated because of the snow cover. In the higher parts of the domain, we adapt the configuration of

Experiment 3 by including only the moulins that are located in non-crevassed areas. In addition, we consider six new moulins, which raises the total number of moulins to 19. These moulins were identified from Fig. 2(c), wherever a supraglacial stream crosses the upper boundary of a crevassed area. The moulin catchments are adjusted as well in order to accommodate the additional moulins, and to exclude the crevassed areas where water drains directly into the subglacial system and does not contribute to the moulin discharge (Fig. 2(d)).

## 3.4 Datasets

### 3.4.1 Study area outline and topography

The TopoToolbox analysis is carried out on a 40 m resolution DEM acquired in 2007 from the SPOT-5 satellite sensor (Bouillon et al., 2006; Korona et al., 2009). Kongsfjord's glacier outlines of 2007 are retrieved from the Randolph Glacier Inventory (König et al., 2013) and are used to shape the model domain. Basal topography is taken from a 150 m resolution

DEM produced by Lindbäck et al. (2018). The glacier outlines are adjusted to avoid zero (and occasionally occurring negative) ice thickness, which would compromise model calculations. Inside the modelled glacier basin, nunataks are also cut out. This model domain, which covers a total surface area of 519 km$^2$, is used for all model runs in this study.

### 3.4.2 Water input

Water supply to the subglacial hydrological system is derived from a surface runoff time series generated by a coupled

surface-energy-balance-snow model that uses downscaled output from the regional climate model HIRLAM as atmospheric forcing (Van Pelt and Kohler, 2015; Van Pelt et al., 2019). The model accounts for subsurface processes such as water storage in the porous firn layer, as well as refreezing/melting of water in the snowpack. Here recharge is the liquid water volume that effectively reaches the glacier bed. More specifically, in the accumulation zone recharge is the amount of liquid water available at the firn-ice transition, which is the sum of percolated rain and surface-generated meltwater minus the





water that is retained in the snow and firn pack (Van Pelt et al., 2016). In the case of bare-ice exposure, such as in the ablation zone during summertime, recharge is the sum of rain and meltwater at the ice surface. The time series covers the period 2003 to 2017 with a 3 hourly time step (subsequently averaged to the 1 day simulation time step) and a 1 km grid resolution.

## 4 Results and analysis

Each Glacier Drainage System model run consisted of a 15 year simulation with one of the four water input configurations described in the previous section. For each model run, we examine the subglacial water pressure (expressed as a fraction of the ice overburden pressure), the water flux in the distributed drainage system (sheet flux) as well as in the channelized drainage system (channel flux), and the absolute value of hydraulic head. We explore the mean seasonal variation of these variables both over the entire glacier basin and at three cross-glacier transects (indicated in Fig. 1(b)). We also study the

spatial distribution of these variables over the whole model domain, as well as along five subglacial flowlines (indicated in Fig. 1(b)). To investigate differences in the subglacial hydrology between the model experiments, we use the simplest scenario (Experiment 1) as a reference against which we compare the other three water input configurations. Finally, we use the most realistic scenario (Experiment 4) to describe the mean spatiotemporal evolution of the subglacial hydrology at the Kongsfjord basin over the study period.

### 4.1 Comparison between experiments

#### 4.1.1 Water input

Between 2003 and 2017, the modelled average melt season lasts approximately from the 1st of June to the 30th of September in all experiments (Fig. 3(a)). In general, despite strong interannual variability, the mean peak in water supply occurs at the end of July. However, input water volumes vary among the four experiments due to different sizes of water contributing

surface area. In Experiments 1–2, the same total water volume ($1.57 \times 10^6$ m$^3$ day$^{-1}$ over 2003–2017) is supplied to the bed because recharge is calculated from the same surface area in both configurations (Fig. 2(a, b)). Experiments 3–4 deliver overall larger water volumes to the bed ($1.76 \times 10^6$ m$^3$ day$^{-1}$ and $2.55 \times 10^6$ m$^3$ day$^{-1}$, respectively, over 2003–2017) than Experiments 1–2 since these scenarios include supraglacial catchments that extend beyond the glacier boundaries (Fig. 2(c, d)). In Experiment 3, the early-summer increase in water input is slightly delayed compared to the other configurations

because this scenario accounts for meltwater only in the higher parts of the glaciers, where surface melting starts later than at lower elevations.





### 4.1.2 Subglacial water pressure

All model experiments exhibit a similar seasonality in responses of the subglacial hydrology (Fig. 3(b)). Each water supply scenario rapidly raises the domain-averaged basal water pressure in the early melt season (June–July), followed by a much slower decline starting at the end of July and lasting through the subsequent winter (Fig. 3(b)). The minimum in water pressure occurs just before the onset of the next melt season. Despite the positive relation between water input and basal pressure on the seasonal scale, the water pressure signal appears smoother than that of the water input on daily timescale, showing less short-term variability. The lowest short-term variability in water pressure is found in the reference scenario (Experiment 1), reflecting a more distributed water input leading to lower amplitude in water pressure fluctuations.

Varying the water input configuration has minimal impact on the wintertime subglacial water pressure (Fig. 4(a–d)), since surface water production outside of the melt season is close to zero. Sensitivity to the water input configuration is highest at the start of the melt season (June), when pressure builds up from low winter levels. Experiment 2 drives the sharpest increase in basal pressure, followed by Experiments 4–1–3 (Fig. 3(b)). Excursions of pressure in Experiment 2 occur in a broad region encompassing all except the two lowermost moulins (Fig. 4(f)), while in Experiment 4, the early-summer pressurisation of the subglacial system is limited to the upper ablation zone (Fig. 4(h)). Experiment 3, despite feeding higher water volumes to the subglacial system, creates lower pressures almost everywhere compared to Experiment 1 (Fig. 4(g)).

Early-summer (June) hydraulic head profiles at each terminal glacier of the Kongsfjord basin represent the height at which water would rise in boreholes drilled to the bed (Fig. 5). This figure supports Fig. 3(b) and Fig. 4, and shows that, at all glaciers, pressure differences between the experiments are minimal at the fronts but increase in the upglacier direction, with Experiment 2 generally being the highest and Experiment 3 always the lowest. In July, pressure biases between the experiments decrease but keep a similar pattern as in June, except that for Experiment 4 the pressure excursions cluster around regions of higher moulin density (Fig. 4(l)). The mid-summer maximum in basal water pressure (~55–65 % of the ice overburden pressure) is quasi simultaneous with the peak in water input (end of July) in all scenarios, except in Experiment 2, which produces a significantly earlier peak in basal pressure (Fig. 3(b)). During the second half of the melt season, Experiment 1 consistently drives higher pressures, whereas Experiments 2–4 have closer agreement (Fig. 3(b)).

Over the mean annual period, largest discrepancies are found between Experiment 1 and Experiment 3, showing highest (17.4 %) and lowest (13.3 %) mean annual water pressures, respectively (Fig. 3(b)). A recurrent pattern in all experiments is that Experiments 2–4 have systematically lower mean water pressures than Experiment 1 in the upper accumulation zones (Fig. 4). This is because in these configurations, meltwater from the upper reaches is supraglacially routed downstream before entering the subglacial system at moulins.



### 4.1.3 Subglacial discharge

We first investigate the partitioning of the subglacial drainage between the distributed sheet and the channels (Fig. 6). In all experiments, the sudden rise in water supply at the start of the melt season rapidly increases the discharge in the distributed sheet without immediately activating channel development. In the upper model domain, the subglacial system remains inefficient throughout the summer, with almost no channel discharge (Fig. 6(a, d)). The sheet responses are relatively similar in all scenarios, except for Experiment 2, which drives an earlier rise but a lower peak in sheet discharge. In the lower model domain, after the initial build-up of the distributed sheet, channels take over and dominate the subglacial drainage from early July to late August (Fig. 6(e, f)). The timing of the switch from inefficient to efficient drainage differs between the water input configurations. With the highest total volume of water input and the fastest response in sheet discharge, Experiment 4 leads, followed by Experiments 2–1–3. In contrast to the distributed system, in which discharge is nearly equal for all experiments, discharge in the channel system varies considerably depending on the water input scenario, with the highest values driven by Experiment 4. Moreover, short-lived pulses in channel discharge are not concurrent in all experiments; however, we focus our analysis away from these features because they likely originate from numerical artefacts. All scenarios agree on that sheet discharge and channel discharge are lower and higher, respectively, at Kronebreen (Fig. 6(f)) than at Kongsbreen (Fig. 6(e)), revealing that subglacial drainage efficiency is not equal in all glaciers at Kongsfjord.

Next, we compare the distribution of subglacial flow paths in the model domain at the peak of channelization for each water supply scenario. As shown in Figure 7, the water input configuration has little influence on the morphology of the channelized system. By the end of August, all experiments produce an arborescent channel network in the lower 15–20 km of the glaciers. Subglacial water is released into the fjord by three major channel trunks that connect the glacier fronts to the upper ablation areas of Kongsbreen North, Kronebreen and Kongsvegen. While subglacial discharge flows in the general downglacier direction, channels do no align with the glacier centrelines, but instead follow the deeper parts of the bed. The complex basal topography of Kongsfjord causes ample meandering in the subglacial flow paths, regardless of the water input configuration. At Kronebreen, a bedrock spur at about 10 km from the calving front splits the main channel into two branches, as highlighted in Fig. 7(d). Regions of shallow ice and adverse slopes in the glacier beds impede channel formation and thus discontinuities occur, in particular in moulin-free Experiment 1 (Fig. 7(a)).

The addition of moulins in Experiments 2–4 induces only little change in the main flow paths structure compared to Experiment 1. However, discrete water recharge produces larger and more continuous channels that reach slightly further inland. The ability of moulins to drive significant channels is nevertheless limited by the amount of surface meltwater captured in their upstream watersheds. In Experiment 4, 11 out of 19 moulins generate channels of flux larger than 1 $m^3\ s^{-1}$, but only seven of these channels reach a sufficient length to connect to a nearby preferential flow path (Fig. 7(d)).

One noticeable difference between the water input scenarios is that subglacial runoff in the Kronebreen/Holtedahlfonna system exits the glacier at different outlets. While in Experiments 1 and 2, subglacial discharge at Kronebreen exits primarily through the north-side frontal outlet (Fig. 7(a, b)), Experiments 3 and 4 drive the southern terminal branch of the Kronebreen



channel to merge with the Kongsvegen channel, leading Kronebreen and Kongsvegen to share the same subglacial outlet on the southern side of their common front. In Experiment 3, most of the subglacial runoff exits through the south-side outlet (Fig. 7(c)), whereas in Experiment 4, the north-side outlet remains connected to the main channel of the Kronebreen/Holtedahlfonna system (Fig. 7(d)).

## 4.2 Seasonal evolution of the subglacial drainage system

Here we use the results of Experiment 4, the most realistic water supply configuration, to describe the subglacial hydrology of Kongsfjord. At the beginning of each melt season (early June), the subglacial hydrological system starts to evolves from its channel-free winter configuration. As rising amounts of surface-generated meltwater penetrate to the glacier beds, the inefficient subglacial drainage system rapidly becomes highly pressurised. Pressure first increases in the ablation zone, where surface melting is highest and crevasses allow immediate transfer of meltwater to the bed (Fig. 8(a)). Water flow at
the ice/bed interface dissipates energy through the melting of ice, thereby opening small channels where melt-opening exceeds creep closure. The first channels (0.2–1.0 m$^3$ s$^{-1}$) appear in late June, at the front of Kongsbreen North and Kronebreen, where water pressure is the highest. Once initiated, channelization rapidly spreads upglacier in the ablation zone, with small channels opening on almost all of the mesh edges.

By mid-July, channels of more significant size (> 1.0 m$^3$ s$^{-1}$) have captured the discharge of the smaller conduits (Fig.
8(b)). This arborescent channel network first establishes in the lowermost part of the domain, before extending towards the upper catchment basins. At the peak of the melt season (July–August), subglacial water pressure varies considerably (0–100 % of the ice overburden pressure) across the Kongsfjord basin, with a domain-wide mean peak value of 59.2 % of the ice overburden pressure. There is a strong inverse correlation between basal water pressure and bedrock topography, the most pressurised regions being the low-lying valleys below the accumulation basins and the subglacial overdeepenings in the
ablation area.

By late August, large channels (> 5 m$^3$ s$^{-1}$) have formed as far as 10–15 km from the glacier fronts (Fig. 8(c)). Although the broad-scale structure of the channelized system remains stable throughout the melt season, channels constantly rearrange to accommodate the short-lived variations in meltwater supply. Many small channels also open in the upper reaches of the domain, mostly at the junction between Isachsenfonna and Holtedahlfonna. These minor channels seem topographically
induced, as they lie across steep bedrock slopes and terminate in the subglacial valley just upstream of the confluence between Kronebreen and Infantfonna. Despite channelization, the subglacial system remains highly pressurised even during the second half of the melting period. In some regions of the glacier basin, basal water pressures in August (Fig. 8(c)) are up to 50 % higher than in June (Fig. 8(a)) for comparable water inputs.

Finally, towards the end of the melt season, the decrease in water supply leads to rapid closure of the subglacial channels,
and the residual water is drained through the distributed system. Channels in the upper reaches are the first to recede, whereas at the glacier termini channels persist until late September.





## 5 Discussion

### 5.1 Model sensitivity to water supply distribution

In this study, we made various assumptions about how meltwater is delivered to the glacier beds. Here we discuss each of
these approximations and how they impact the development of the subglacial drainage system in Kongsfjord.

Our reference experiment (Experiment 1) assumes direct and immediate surface-to-bed water transfer over the entire model domain, by considering each node of the mesh as a sink for the local meltwater. Despite being commonly used in most models of subglacial hydrology, the assumption of perfect vertical permeability may only be appropriate in heavily crevassed glacier zones, such as the terminal part of Kronebreen. This approximation overestimates the amount of water
supply to large parts of the glacier bed, in particular at higher elevations, but underestimates input in areas where recharge is concentrated at moulins. Accordingly, Experiment 1 drives higher water pressures in the upper accumulation zone (Fig. 4) and highest mean annual pressures overall compared to the other scenarios (Fig. 3(b)).

In Experiment 2, the domain-wide total recharge is evenly distributed among 13 moulins instead of being directly fed to the subglacial hydrological system. The assumption of equal recharge at moulins is unrealistic as it implies that meltwater
produced at lower elevations is injected into moulins at higher elevations. Results from this experiment can be easily compared with those of Experiment 1, since both scenarios apply the same water volume to the subglacial system (Fig. 3(a)). Experiment 2 drives an earlier but lower peak in subglacial water pressure than Experiment 1 (Fig. 3(b)), indicating that more localised recharge favours earlier channel formation through locally higher subglacial discharge (Fig. 6(a–c)).

In Experiment 3, the input to each moulin is collected from their upstream catchments, generating a slightly larger total
volume of water supply than in Experiments 1–2 (Fig. 3(a)). Yet paradoxically, Experiment 3 produces overall lower water pressures than Experiment 1 (Fig. 3(b), Fig. 4(g, k)) and poor channelization in the ablation zone (Fig. 6(e, f)) compared to the other scenarios. We attribute this counterintuitive result to the water supply configuration that does not account for recharge in the ablation zone, where meltwater production is highest.

Experiment 4 aims at the most realistic representation of the actual supraglacial hydrology of the Kongsfjord basin by
considering supraglacial streams, supraglacial lakes, moulins and crevasses. The total water volume delivered to the bed is higher than in the other experiments due to the different drainage catchment definition (Fig. 3(a)). Similar to Experiment 2, this scenario leads to both earlier and higher subglacial discharge compared to Experiments 1 and 3 (Fig. 6(a–c)), thus driving higher channelization in the ablation zone (Fig. 6(b, c)).

Overall, varying the water input configuration does not drastically influence the seasonal evolution of the subglacial
drainage system. All experiments lead to similar channel networks by the end of the summer (Fig. 7), albeit differences are noticeable in the transition period from an inefficient winter state towards the more efficient summer state (Fig. 6). More specifically, Experiments 2 and 4 introduce differences on how the subglacial drainage system copes with early-summer increase in water supply, leading to significantly higher water pressures at the onset of the melt season (Fig. 4(f, h)) and





faster channelization in the ablation zone (Fig. 6(b, c, e, f)), which subsequently causes lower pressures in the ablation zone
than in Experiment 1 (Fig. 4(j, l)).

## 5.2 Low efficiency of the subglacial drainage system

In contrast to some previous studies (e.g. Schoof, 2010; Chandler et al., 2013), we find no support that channels are able to
substantially increase the efficiency of the subglacial drainage system, as water pressures remain high during the entire
melting period in all of our experiments (Fig. 3(b)). As depicted in Figure 9, the subglacial water pressure displays annual
hysteresis under variations in water input. Experiments 1–3–4 exhibit strong counter-clockwise hysteresis (Fig. 9(a, c, d)),
meaning that for comparable water inputs, water pressure is significantly higher during falling recharge at the end of the melt
season than during rising recharge at the start of the melt season. This indicates that water pressure continuously builds up
during the melt season and only slowly decreases afterwards. As also shown in Figure 3(b), the lag between water pressure
changes in response to water input changes implies that, by the end of the melt season (late September), the subglacial
hydrological system has not returned to its previous wintertime configuration. This supports our interpretation that channels
contribute only very little to the modulation of subglacial water pressure in our study area. For Experiment 2, the relationship
between water input and water pressure is more complex. At higher water inputs (above 60 m$^3$ s$^{-1}$), the hysteresis loop turns
clockwise (Fig. 9(b)), indicating that the subglacial drainage system becomes more efficient with increasing recharge. This
change of direction in hysteresis between late June and mid-August indicates that subglacial channels are able to exert some
control, although limited, on basal water pressure. Moreover, in our most realistic water supply scenario (Experiment 4),
high water pressures (> 50 % of the ice overburden pressure) until late summer over large parts of the model domain (Fig.
8(c)) suggest that channelization only occurs at high pressurisation of the subglacial drainage system. Channels of significant
size (> 1 m$^3$ s$^{-1}$) open but are restricted to the lower part of the model domain, and the presence of small channels in many
regions of the glacier bed (Fig. 8(b, c)) also points to the predominance of a distributed drainage system, rather than a true
channelized system. We identify two factors behind the low efficiency of the channels in our study area.

First, our model results show that the spatial distribution of channels is primarily controlled by the bed topography,
regardless of the water input configuration. Channels form along steep basal slopes and basal overdeepenings, where
gradients of hydraulic potential are highest. All of our experiments produce similar spatial patterns in the summertime
drainage efficiency, with the largest subglacial channel occurring underneath the Kronebreen/Holtedahlfonna system. The
location of the modelled channel network and its outlets in the lower part of Kronebreen is consistent with findings of How
et al. (2017), who used a simpler approach for determining subglacial flow paths, assuming a static hydraulic potential. The
small surface gradients that characterise our model domain also drive the high pressurisation of the channels. Subglacial
channels open through melting of the overlying ice driven by turbulent heat dissipation, which requires steep downglacier
bed and surface slopes that are absent in our study area. Hence, higher water pressures are required to counteract creep
closure and maintain the channels open, even in areas that receive high water input from moulins. The finding that subglacial





channels do not play an important role in regulating subglacial water pressure at such low sloping glaciers has been reported in other studies applied to the Greenland Ice Sheet (Meierbachtol et al., 2013; Moon et al., 2014; Dow et al., 2015).

Second, arctic glaciers such as those of the Kongsfjord basin experience shorter melt seasons than mid- and low-latitude glaciers, and thus receive overall less water supply from the surface, which also disadvantages channelization. In particular, minimal wintertime water input due to generally low temperatures in the Arctic and limited recharge from episodic warm spells is unable to sustain year-round subglacial channels. Compared to overwintered channels that more readily grow to accommodate meltwater input, first season channels lack the capacity to evacuate large volumes of water injected to glacier beds. Poinar et al. (2019) suggested that high-evelation firn aquifers can provide steady water input to the subglacial system outside of the melt season, thereby facilitating subglacial channel formation and winter-persistence, even in low sloping areas. Evidence of a perennial firn aquifer was revealed by ice-penetrating radar and GPS observations in the upper reaches of Holtedahlfonna (Christianson et al., 2015). This aquifer is recharged by downward percolation of summertime meltwater, and discharge occurs through downhill flow in the winter. Deep firn water storage is accounted for in the model that provided the input time series to our subglacial hydrology model (Van Pelt et al., 2019), but does not permit considerable wintertime contribution to recharge in absence of supply from the surface.

## 5.3 Implications for glacier sliding

Studies of glacier motion suggest that ice acceleration typically occurs after the onset of the melt season, when water input to the ice/bed interface is rising faster than the subglacial drainage system can accommodate, driving an increase in basal water pressure, ice-bed separation, and sliding. The sensitivity of ice motion to water input usually decreases over the melt season as the subglacial drainage system evolves from an inefficient distributed to efficient channelized configuration, thereby gradually lowering basal water pressures despite increasing surface melt, and thus causing ice deceleration in late summer (Davison et al., 2019).

Instead, our model results show that basal water pressures remain high during the entire summer, which we attribute to low hydraulic gradients that in turn disfavour the development of efficient channels. We thence deduce that substantial sliding likely occurs at most of the Kongsfjord glaciers over the entire melt season, and that sliding rates remain high even in the late summer, which is consistent with observations by Schellenberger et al. (2015). Limited channelization, by maintaining high water pressures in the subglacial system, further enhances sensitivity of sliding to basal recharge variability.

As shown in Figure 4, the sensitivity of the modelled subglacial hydrology to the water supply configuration is highest at the start of the melt season, when uniform water recharge is locally provided at moulins only (Experiment 2), as well as in the upper accumulation zone, depending on whether water is supplied to the subglacial hydrological system (Experiment 1) or not (Experiments 2–4). In an ice flow model forced by a subglacial hydrology model such as that used in this study, both Experiments 1 and 2 would cause unrealistically high sliding in the upper accumulation zone (Experiment 1) and at higher-





elevation moulins (Experiment 2), as these regions receive too much water. Conversely, the absence of water recharge in the ablation zone (Experiments 2 and 3), by driving only moderate pressurisation of the subglacial system, would underestimate

the rate of sliding at lower elevations. Finally, the combination of both continuous water recharge in the ablation zone and discrete water recharge in the accumulation zone (Experiment 4) drives the highest early-summer water pressures in the lower reaches of the glacier basin, which is in accordance with the observation of large fluctuations in seasonal velocity in this area (Schellenberger et al., 2015).

## 5.4 Model limitations and future work

One limitation of this study is the lack of coupling between subglacial hydrology and ice dynamics. Given the regional scale of our experiments, running the model in an uncoupled version allowed us to keep the computational cost of our study at a reasonable level. Therefore, we have primarily focused our discussion on the behaviour of the subglacial hydrological system under different melt input configurations, and have only cautiously discussed its likely impacts on sliding. Implementing a two-way coupled model would be relevant for future studies seeking to investigate in more detail the

feedbacks between subglacial hydrology and ice dynamics.

Another limitation is that our modelling approach ignores some aspects of the glacial hydrological system. We assume that the location of surface-to-bed connections such as moulins and crevasses remains unchanged during the 15 year period of our simulations. In reality, supraglacial hydrology is a dynamic system that is governed by variations in ice velocity and in surface mass balance. Glacier acceleration enhances crevassing, thus increasing the area of direct meltwater transfer to the

ice/bed interface. Intense melt events not only promote opening of new moulins through melt-enlargement of existing crevasses, but also drive hydrofracturing, a mechanism through which water-filled crevasses or supraglacial meltwater ponds abruptly propagate downwards (Alley et al., 2005, van der Veen, 2007; Das et al., 2008). Supraglacial lake drainage is a widespread mechanism for moulin activation in the upper ablation zone, and hence may contribute significantly to subglacial reorganisation (Bingham et al., 2003; Andrews et al., 2018; Hoffman et al., 2018). Firn aquifers lying upstream of crevasse

fields can also drive hydrofracturing (Poinar et al., 2017). Consequently, neglecting the spatiotemporal evolution of crevasses, moulins and lake drainage events in our model is definitely a limiting factor to the accurate simulation of water recharge to glacier beds. This limitation could be overcome by using a spatially distributed model of supraglacial hydrology that would predict both the location and the timing of new surface-to-bed hydrological connections for given surface melt scenarios (e.g. Clason et al., 2012; 2015; Koziol et al., 2017).

## 490  6 Conclusions

The results of our modelling indicate that discrete basal water recharge at moulins (Experiments 2 and 4) leads to higher water pressures at the beginning of the melt season, exceeding those simulated when assuming spatially continuous, direct





surface-to-bed transfer (Experiment 1). However, the more localised recharge also favours a faster evolution of subglacial channels, with an opposite effect that quickly dominates such that the moulin configurations result in lower water pressures
during the remaining of the melt season. The modelled timing at which sliding would start to occur is therefore sensitive to moulin distribution and basal recharge. In this regard, we conclude that there is value in mapping of supraglacial hydrology features such as streams, crevasses, lakes and moulins in order to improve constraining of recharge to glacier beds and accurately determine the distribution of meltwater input into moulins.

Nevertheless, our model results suggest that the modelled subglacial hydrology in our study area is overall poorly
sensitive to the water input configuration. We find that both the location and the size of subglacial channels are governed largely by basal topography, regardless of how and where water enters the glacier beds. We attribute this to the gentle slopes of the glaciers that do not support strong hydraulic gradients and to the short duration of the high arctic melt season with limited amounts of meltwater injected to the subglacial hydrological system. Therefore, subglacial channels are unable to substantially decrease the summertime overall basal pressure, implying that sliding is directly linked to spatiotemporal
variability in meltwater recharge, and that sliding likely occurs at the Kongsfjord basin during the entire melting period.

The outcomes of our study raise implications for the future of ice dynamics in a warming climate and can be generalised to most Svalbard tidewater glaciers, which have relatively flat surfaces and which experience typically less water supply to their beds due to their high latitude. The inability of these glaciers to build an efficient drainage system implies that they are likely sensitive to long-term trends in meltwater supply variability, contrary to what would be expected in a more standard
configuration (Schoof, 2010).

**Data availability**

The digital elevation model can be accessed at https://doi.org/10.21334/npolar.2014.dce53a47 (Norwegian Polar Institute, 2014). The glacier and land masks were constructed from glacier outlines, which are available at https://doi.org/10.21334/npolar.2013.89f430f8 (König et al., 2013). The subglacial topography and bathymetry of Kongsfjorden is available at
https://doi.org/10.21334/npolar.2017.702ca4a7 (Lindbäck et al. 2018). The modelled water recharge dataset has been prepared and shared by Ward Van Pelt, and can also be accessed in the following repository: https://doi.org/10.6084/m9.figshare.7836530.v1 (Van Pelt et al. 2019). Crevasse zones and moulin locations have been mapped from commercially available, licenced satellite imagery (CNES (2014) and Airbus DS (2014), all rights reserved. Commercial uses forbidden). The TopoToolbox is available at https://github.com/wschwanghart/topotoolbox (Schwanghart and Scherler, 2014). The subglacial hydrology model GlaDS is freely
available as part of the Elmer/Ice model (Gagliardini and Werder, 2018).

**Author contributions**

All three authors designed the research idea, CS performed the simulations and the main analysis. CS wrote the paper with contributions from all co-authors.



**Competing interests**

The authors declare that they have no conflict of interest.

**Acknowledgements**

The authors would like to thank Ward Van Pelt for providing the water recharge time series used as input to the subglacial hydrology model. We are grateful to Jack Kohler and Chris Nuth for their support in obtaining and processing digital elevation models and satellite imagery. Chloé Scholzen would like to thank Olivier Gagliardini for fruitful discussions and assistance during her research stay at IGE, CNRS.

**Financial support**

This research has been supported by the Norwegian Ministry of Education and Research who granted a PhD fellowship to CS, and the Research Council of Norway through the project MAMMAMIA (grant no. 301837).

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





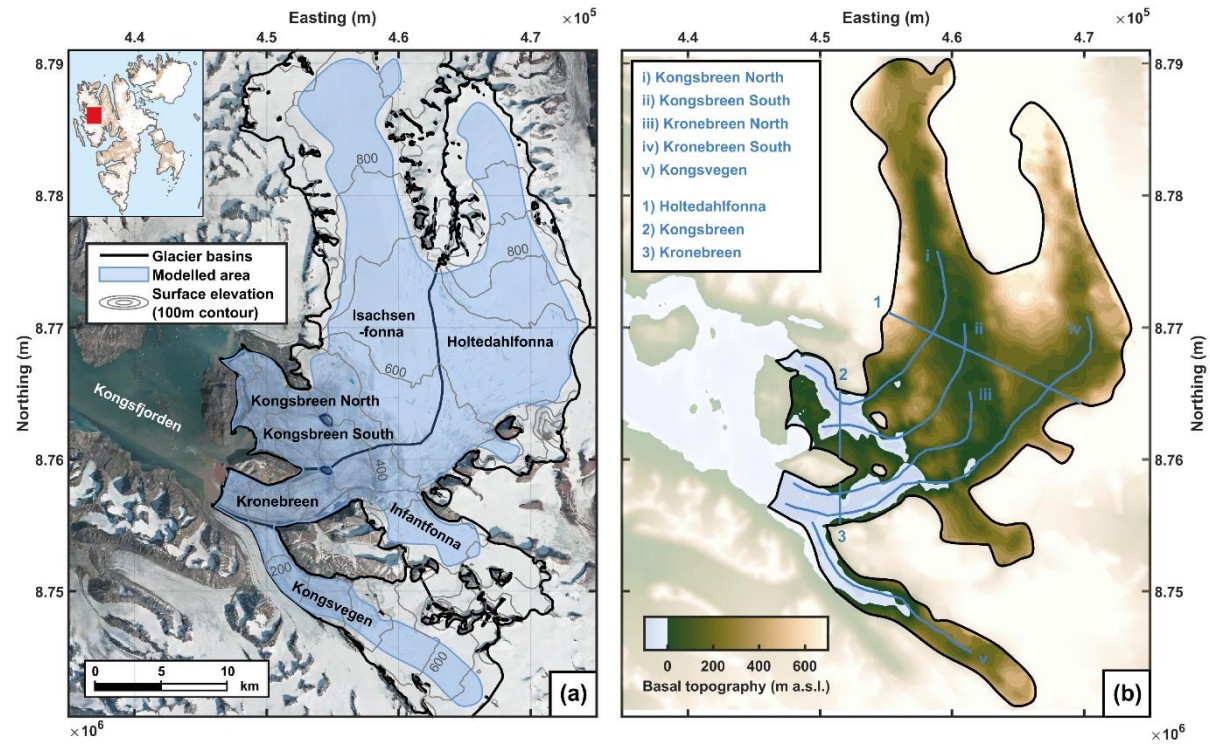

**Figure 1.** Kongsfjord basin with black line outlining the model domain. **(a)** Surface topography (Lindbäck et al., 2018) (background image: https://toposvalbard.npolar.no/, Norwegian Polar Institute), **(b)** Basal topography (Lindbäck et al., 2018) with (i–v) subglacial flowlines and (1–3) cross-glacier transects defined for analysing the model outputs.

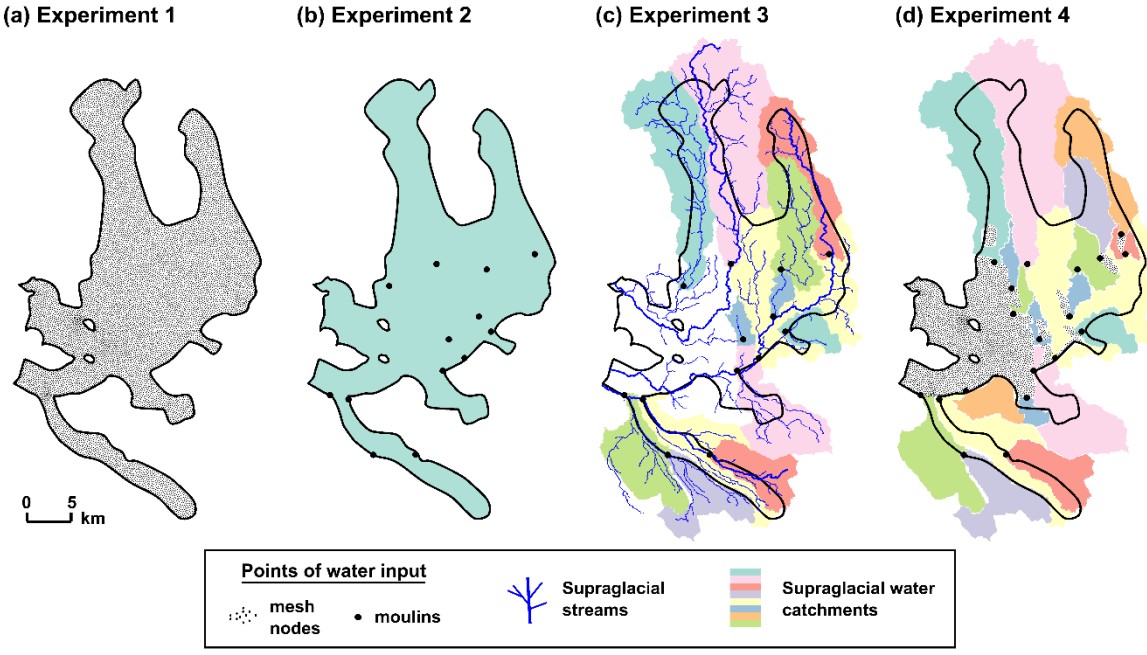

**Figure 2.** Water input distribution at the Kongsfjord basin for each experiment. Shown in black are the points of water recharge to the
subglacial hydrological system, i.e. mesh nodes for **(a, d)** and moulins for **(b–d)**. The coloured areas are the supraglacial water catchments
associated to the moulins. Subplot **(c)** also shows the TopoToolbox generated supraglacial stream network in blue.





**Figure 3.** Mean (2003–2017) annual **(a)** water input and **(b)** basal water pressure averaged over the whole model domain for each
experiment. The shaded area is the standard deviation showing the interannual variability of water input and water pressure for each
experiment.





**Figure 4.** Mean (2003–2017) basal water pressure modelled by Experiment 1 in **(a)** late winter (April), **(e)** early summer (June), and **(i)** mid-summer (July). Biases (differences to reference Experiment 1) in mean (2003–2017) basal water pressure in **(b,c,d)** late winter (April), **(f,g,h)** early summer (June), and **(j,k,l)** mid-summer (July). Basal water pressure is expressed as a fraction of the ice overburden pressure. The black dots mark the moulin locations.




**Figure 5.** Mean (2003–2017) early melt season (June) hydraulic head modelled by each experiment at **(a)** Kongsbreen North, **(b)** Kongsbreen South, **(c)** Kronebreen North, **(d)** Kronebreen South, and **(e)** Kongsvegen. Hydraulic head is expressed in absolute values (m a.s.l.). Flowlines locations are shown in Fig. 1(b).


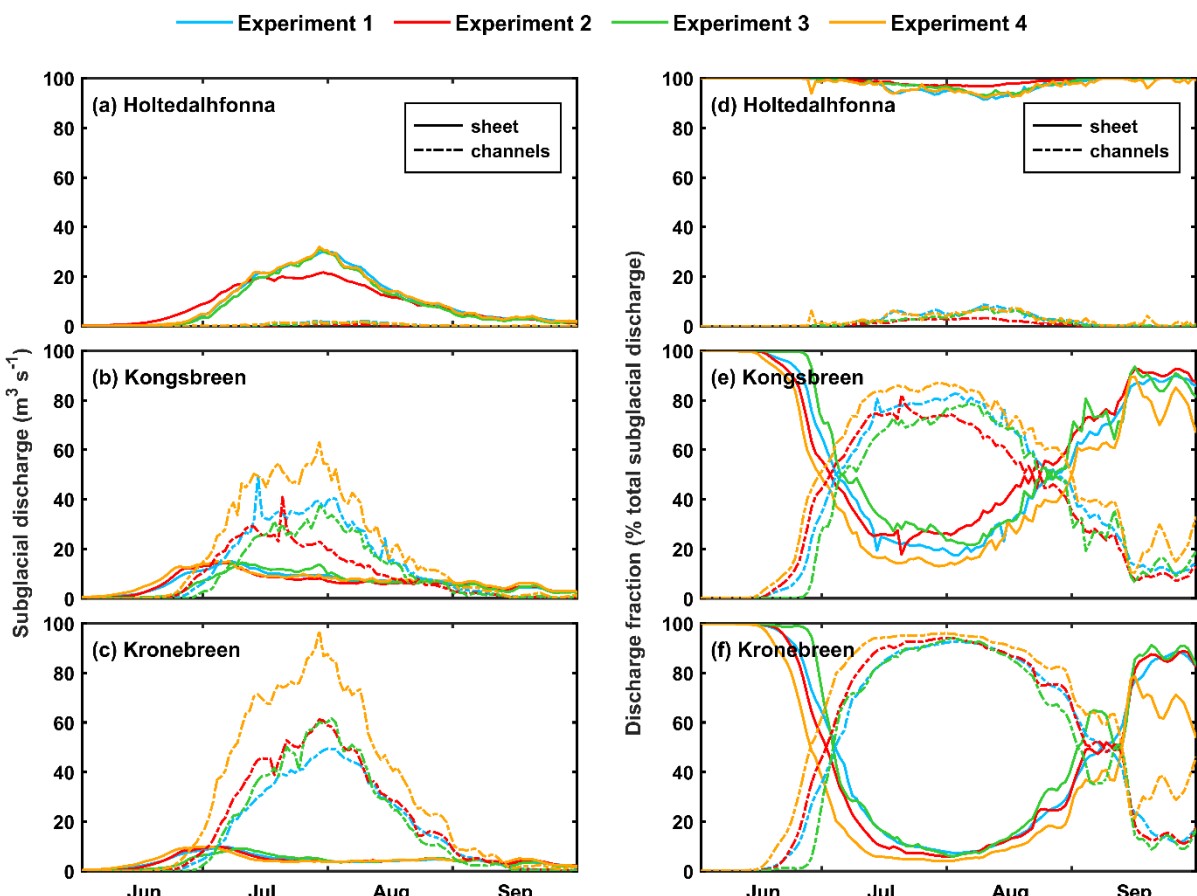

**Figure 6. Left panel:** Mean (2003–2017) summertime (June–September) sheet discharge and channel discharge modelled by each experiment across **(a)** Holtedahlfonna, **(b)** Kongsbreen and **(c)** Kronebreen. Sheet discharge is the cross-glacier width integral of the x

component of the nodal sheet discharge. Channel discharge is the sum of discharge from all channel edges intersecting the cross-glacier section. Discharge is expressed in absolute values (m³ s⁻¹). **Right panel:** Mean (2003–2017) summertime (June–September) sheet discharge and channel discharge modelled by each experiment across **(d)** Holtedahlfonna, **(e)** Kongsbreen and **(f)** Kronebreen. Here, discharge in the sheet and in the channels is expressed as a fraction of the total discharge (the sum of the discharge in the sheet and in the channels). Cross-glacier transects locations are shown in Fig. 1(b).





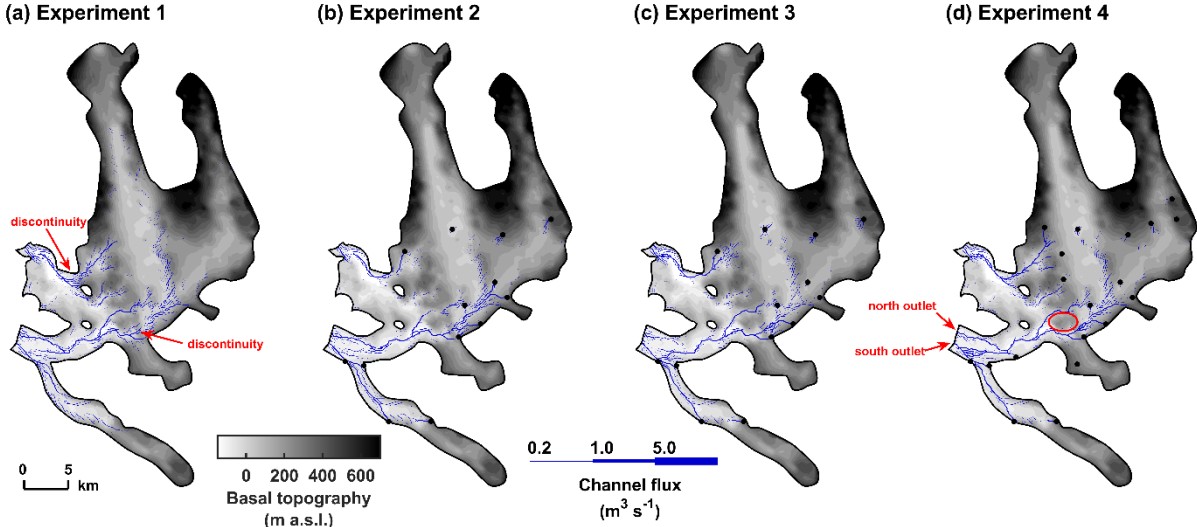

**Figure 7.** Summer snapshot (20th of August 2013) of subglacial channels distribution modelled by each experiment, with subglacial topography in greyscale. The year (2013) is chosen arbitrarily but is representative of the entire period. We use 1 m³ s⁻¹ as an arbitrary
threshold flux to discriminate regions of significant channel size. Discontinuities in the channel network due to basal adverse slopes are marked in red in Experiment 1 **(a)**. Bedrock spur is circled and frontal outlets to subglacial runoff at Kronebreen/Kongsvegen are marked in red in Experiment 4 **(d)**. The black dots mark the moulin locations.





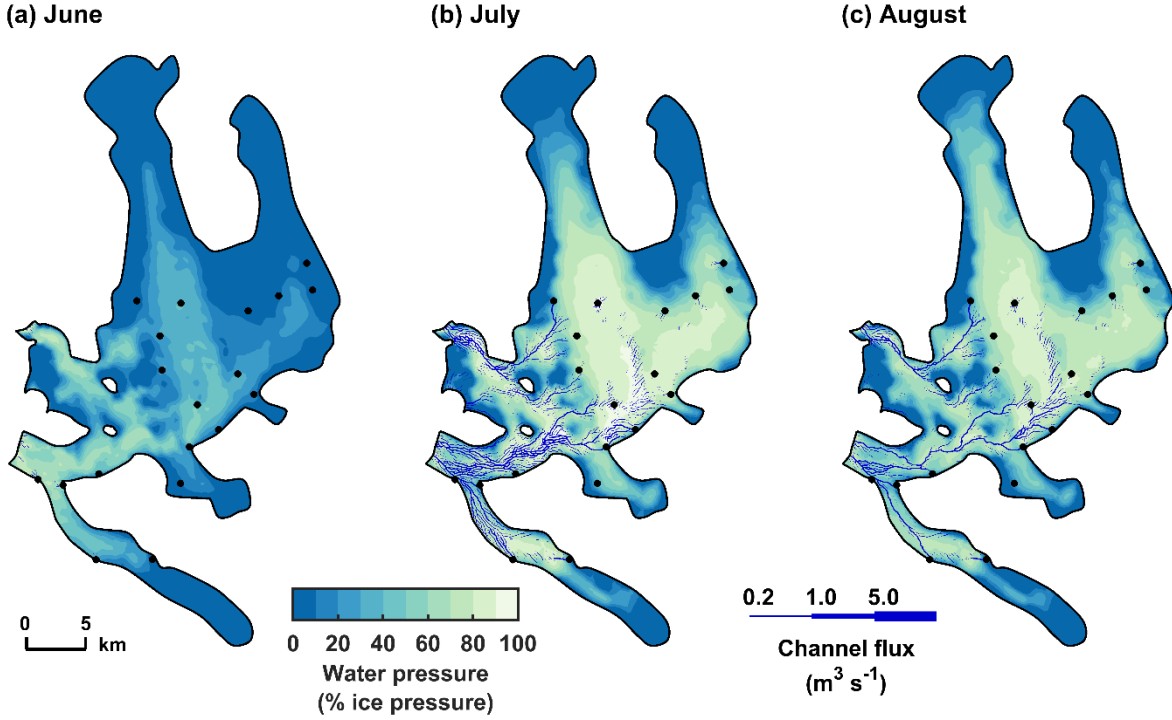


**Figure 8.** Mean (2003–2017) basal water pressure and channel discharge modelled by Experiment 4 in **(a)** early summer (June), **(b)** mid-summer (July), and **(c)** late summer (August). Snapshots of channel discharge are taken on (a) the 20th of June 2013, (b) the 20th of July 2013, and (c) the 20th of August 2013. The year (2013) is chosen arbitrarily but is representative of the entire period. Basal water pressure is expressed as a fraction of the ice overburden pressure. We use 1 m³ s⁻¹ as an arbitrary threshold flux to discriminate regions of

significant channel size. The black dots mark the moulins locations.





**Figure 9.** Relationship between water input and basal water pressure averaged over the whole model domain for each experiment during
the mean (2003–2017) annual period. Clockwise hysteresis loop in Experiment 2 (b) between late June and mid-August is marked in red.
Water pressure is expressed as a fraction of the ice overburden pressure (%).



| Description | Symbol | Value | Literature value | Units |
|---|---|---|---|---|
| Pressure melt coefficient | $c_t$ | $7.5 \times 10^{-8}$ | $7.5 \times 10^{-8}$ | K Pa$^{-1}$ |
| Heat capacity of water | $c_w$ | 4,220 | 4,220 | J kg$^{-1}$ K$^{-1}$ |
| First sheet flow exponent | $\alpha_s$ | 5/4 | 5/4 | - |
| Second sheet flow exponent | $\beta_s$ | 3/2 | 3/2 | - |
| First channel flow exponent | $\alpha_c$ | 5/4 | 5/4 | - |
| Second channel flow exponent | $\beta_c$ | 3/2 | 3/2 | - |
| Sheet conductivity | $k_s$ | 0.005 | 0.01 | m$^{7/4}$ kg$^{-1/2}$ |
| Channel conductivity | $k_c$ | 0.5 | 0.1 | m$^{3/2}$ kg$^{-1/2}$ |
| Ice flow constant cavities | $A_s$ | $6.8 \times 10^{-24}$ | $6.8 \times 10^{-24}$ | Pa$^{-3}$ s$^{-1}$ |
| Ice flow constant channels | $A_c$ | $6.8 \times 10^{-24}$ | $6.8 \times 10^{-24}$ | Pa$^{-3}$ s$^{-1}$ |
| Sheet width below channel | $l_c$ | 5 | 2 | m |
| Cavity spacing | $l_r$ | 2 | 2 | m |
| Bedrock bump height | $h_r$ | 0.1 | 0.1 | m |
| Englacial void ratio | $e_v$ | $10^{-4}$ | $10^{-4}$ | - |
| Moulin cross-sectional area | $A_m$ | 4 | 4 | m$^2$ |
| Basal sliding speed | $u_b$ | $10^{-5}$ | $10^{-6}$ | m s$^{-1}$ |


**Table 1.** Parameters and values used in GlaDS for all model runs in this study. Grey highlighted parameter values are those that are not taken from literature. Literature values are taken from Werder et al. (2013).