# Peer review of "Sensitivity of subglacial drainage to water supply distribution at the Kongsfjord basin, Svalbard"

_The Cryosphere, 2020_

## Referee Comment (RC1) · Anonymous Referee #1 · 21 Dec 2020

%% General comments

This paper studies the dependence of the subglacial drainage system at the Kongsfjord basin on the characteristics of the supraglacial hydrological system. The implications of this study extend beyond the Kongsfjord basin as the dynamical behaviour of subglacial hydrological systems in general is not well understood. This is a particularly important question near the margins of ice-sheets because the structure of the subglacial drainage system will directly effect the response of the sheet to increased surface melting.

The paper presents 4 experiments; each with a different supraglacial drainage con-

figuration to supply meltwater into the subglacial drainage system. A general trend between experiments is that more intense meltwater input into the subglacial system will result in more chanelisation; however, not enough to develop long-term efficient subglacial drainage.

It seems like some of the findings are in line with those of Koziol and Arnold 2018 (modelling seasonal meltwater forcing on the velocity of land-terminating margins of the Greenland ice-sheet). In this paper they consider how a subglacial drainage system might contribute to decadal-timescale slowdown of ice in the ablation zone. One theory is that if a more efficient drainage system develops in the summer, the effective pressure throughout the winter will be higher (i.e. the water pressure will be lower), leading to slow down. Your results (figure 3b) seem to suggest that supraglacial-meltwater configurations that produce more efficient drainage (in summer) lead to lower water pressures in winter (as Koziol and Arnold suggest).

You conclude that the resulting subglacial drainage systems that evolve are inefficient. I think here efficiency refers to how the water pressure responds to meltwater input. It seems though (I may be wrong) that the drainage systems are efficient enough to entirely drain the meltwater before the next melt season. Thus, we have the "worst of both worlds" in that the drainage system is inefficient enough for a spike in water pressure at the start of every melt season, but efficient enough to completely drain and revert to it's original (most inefficient) state before the start of the next melt season.

Overall though, without a sustained efficient drainage system, melt added during the summer melt season will drive higher water pressures, which may lead to faster ice-speeds.

I found that, stylistically, this paper was easy to read and well written.

%% Specific comments

A key result is that efficient drainage systems do not develop, which means that the

summertime water pressures can be large. My question would be how much this depends on the choices of conductivity values?

It was mentioned that these values were chosen to "facilitate chanelisation and maximise the influence of meltwater on subglacial drainage efficiency", where it seems efficiency is inversely related to peak water pressure. This meant taking a sheet conductivity smaller than the literature value, and a channel conductivity larger than the literature value. I can see that this will increase both the channel flux and therefore the channel opening rate. Is it possible that lower conductivity terms could lead to more efficient drainage in the long run? Namely, if higher water pressures are sustained throughout winter, the channels may not full close. It doesn't seem like your simulations predict persistent channels throughout the year. I notice that you do address the idea of "sustained year-round subglacial channels" and why this doesn't happen at the Kongsfjord basin. That is, less overall meltwater input and particularly cold winter temperatures. It may be that I'm completely wrong here, I was just wondering if you had any thoughts on the issue of how strongly connected your results are to your choice of conductivity parameters, and can you be sure that you are "maximising the influence of meltwater on subglacial drainage efficiency"?

A final question that, again I just wonder if you have any thoughts on, is what is the missing ingredient that would produce more efficient drainage systems at the Kongsfjord basin? Is it simply that the meltwater supplied to the subglacial hydrology system is too small, or that the parameter values are insufficient. If, for example, you could choose conductivity values that would give "efficient drainage", are they too far away from the physical values to be plausible?

To summarise my general question is how much of an influence do your parameter choices (hydraulic conductivities) have on the qualitative behaviour of your results?

%% Technical corrections

Line 69) Colon misuse, I think you can just remove it.

[Figure]

Line 70) I think "approximations on" should be something more like "approximations about" or "approximations of".

Line 98) Should be a double hyphen (en dash) between "balance" and "snow".

Line 151) Colon misuse (clause before colon should be independent). I also think semi-colons should be used to separate list items here because there is internal grammar in each of the items (commas).

Line 323) "channels do no align"-> "channels do not align".

Line 434 + 502) Capitalisation of "arctic" (-> "Arctic").

Line 434) "disadvantages chanelization" seems a bit awkward, maybe "inhibits chanelization" or "prevents chanelization" (I think this one is just personal taste so happy for it to be ignored).

---

## Referee Comment (RC2) · Christine Dow (Referee) · 6 Jan 2021

This paper using the GlaDS subglacial hydrology model to examine the importance of channels and water pressure for ice dynamics of Svalbard glaciers. The study site has a good volume of support data for this type of project and the authors have tackled their research questions by running 4 experiments to test whether the location (and to some extent the volume) of water input has an impact on channelization and general drainage system development. The primary findings are that water pressure increases until the middle of the summer along with channelization and then both the pressure and channels diminish as the surface water inputs reduce in volume. Over winter the

pressure then drops down to a lower level. I appreciate the application of hydrology models to these glaciers which differ from other regions in that they have a low surface slope and less water input than many lower latitude glaciers and therefore can illuminate different controls on ice dynamics for these types of polythermal glaciers.

MAIN POINTS

This is a well written paper and there is some interesting analysis about the hydrology. I do, unfortunately, have a major concern, which is that the subglacial water pressure is far too low. It drops down to almost 0% of overburden during winter which is very unrealistic and then in the summer the mean pressure is less than 80%. From Figure 8, it looks like only a very small region of your domain gets to overburden pressure with the rest significantly lower. As a reference, boreholes that hit efficient systems often have pressure varying $\sim$ 20-60% overburden and that is considered low pressure. The distributed system should have high pressure, which would be anything above about 80% overburden.

The reason that you have the seasonal and channelization behaviour that you see is that the model is spending most of the season building up to a background level of pressurization, which you would normally assume it would already have at the beginning of the season. Instead, with the spring event, water inputs should be into a system already close to overburden pressure. The rapid ice acceleration during this time is because often the basal system will increase to pressures above overburden, hydraulically jacking up the ice and allowing fast flow. I notice that you don't note in the manuscript how you spin up the model. For GlaDS (and many other models), you have to have a spin-up period so that the system can adjust to the background inputs, which in this case should be whatever basal water is available. You then have to make sure your chosen parameters allow as realistic a system as possible for when you initiate you seasonal inputs. This is why sensitivity tests are often used to assess the variations that parameters will have on the system and, with GlaDS, the two most important parameters to test are the sheet and channel conductivities. Set either too high and

the system won't pressurize and is unrealistic. Set either too low and the model will break because the system will become too pressurized. From those sensitivity tests you then have a range of applicable conductivity values to use as a starting off point for your four experiments. If you have a look at the GlaDS literature for the Antarctic (Dow et al, 2018; Dow et al, 2020) and from Greenland (Poinar et al, 2019; Cook et al, 2020), you'll see that the sheet and channels conductivity values used are much lower (<1e-4 and <0.1, respectively) than have been applied in your study, which explains the low pressures throughout your domain.

GlaDS also tends to have some issues with over-winter pressures if a spatially and temporally uniform basal sliding speed is used. This is because the basal sliding is applied in the cavity opening term. I would recommend taking the basal sliding rate as a percentage of the surface sliding rate to get the spatial variability, and then adjust this temporally using records of summer vs. winter velocity (if you have them). The latter doesn't have to be high temporal resolution but a lower sliding speed in winter causing less cavity opening will allow the system to repressurise (that's assuming that winter sliding speeds are lower than in the spring).

My final main point is that, on the assumption you have access to ice surface velocities for the region, that is the best way to test whether the model is correctly representing your study region. Even a spatially averaged mean velocity should generally match the mean water pressure records that you show in Figure 3b. If these have the same pattern it would make your arguments about the subglacial system evolution stronger.

SPECIFIC COMMENTS

61 - why would the hydraulic potential minimum seed channels normally?

157- "which participated". Also what do you mean by distributed model in this sentence?

178 – when you say HP 'set to zero' how do you apply that? As tidewater glaciers the

outlet boundary condition would be best set at overburden but it's not clear if you do this.

197 and 373 – need some references for this statement. Most recent subglacial hydrology studies do use moulin inputs.

206 – why only keep 10? How does this turn into 13?

265 – It would be useful to know what that input is in m3/s in addition to the cumulative input for the total catchment that you state.

270 – why are the moulins only higher up (I may have missed this)?

289 – state what output result it supports rather than what figure

314 – what kind of numerical artefacts? Why would these occur?

Data availability – model outputs not provided.

Figure 1 - more detail needed for your below-sea-level elevations in panel b)

Figure 2- you have a lot of moulins on boundary points. That might cause problems if you reduce the conductivity to get the system closer to overburden.

Figure 6 and 9 – These discharge fraction and hysteresis diagrams are a nice way to examine your results.

Table 1 – basal sliding speed would be better stated in m a-1 and match what you say in the text.

REFERENCES

Cook, S.J., Christoffersen, P., Todd, J., Slater, D. and Chauché, N., 2020. Coupled modelling of subglacial hydrology and calving-front melting at Store Glacier, West Greenland. The Cryosphere.14,905-924

Dow, C.F., Werder, M.A., Babonis, G., Nowicki, S., Walker, R.T., Csathó, B., Morlighem,

M., 2018. Dynamics of active subglacial lakes in Recovery Ice Stream. J. Geo-phys. Res., Earth Surf.123, 837– 850.

Dow, C.F., McCormack, F.S., Young, D.A., Greenbaum, J.S., Roberts, J.L. and Blanken-ship, D.D., 2020. Totten Glacier subglacial hydrology determined from geophysics and modeling. Earth and Planetary Science Letters, 531, p.115961.

Poinar, K., Dow, C.F., Andrews, L.C., 2019. Long-term support of an active sub-glacial hydrologic system in southeast Greenland by firn aquifers. Geophys. Res. Lett.46, 4772–4781

---

## Author Comment (AC1) · 23 Feb 2021

General comments

**(1) It seems like some of the findings are in line with those of Koziol and Arnold 2018 (modelling seasonal meltwater forcing on the velocity of land-terminating margins of the Greenland ice-sheet). In this paper they consider how a subglacial drainage system might contribute to decadal-timescale slowdown of ice in the ablation zone. One theory is that if a more efficient drainage system develops in the summer, the effective pressure throughout the winter will be higher (i.e. the water pressure will be lower), leading to slow down. Your results (figure 3b) seem to suggest that supraglacial-meltwater configurations that produce more efficient drainage (in summer) lead to lower water pressures in winter (as Koziol and Arnold suggest).**

*Response:* We thank the reviewer for mentioning this paper. Our results are indeed in accordance with those of Koziol and Arnold (2018) in that the configurations that produce more efficient drainage in summer lead to lower water pressures in winter. This is mainly visible in Experiment 1, which, compared to the other three configurations, leads to higher water pressures during both summer and winter due to less channelization. This paper has been appropriately referenced in the revised manuscript to highlight this point.

**(2) You conclude that the resulting subglacial drainage systems that evolve are inefficient. I think here efficiency refers to how the water pressure responds to meltwater input. It seems though (I may be wrong) that the drainage systems are efficient enough to entirely drain the meltwater before the next melt season. Thus, we have the "worst of both worlds" in that the drainage system is inefficient enough for a spike in water pressure at the start of every melt season, but efficient enough to completely drain and revert to it's original (most inefficient) state before the start of the next melt season.**

*Response:* Following the comments of the reviewers on the possible impact of a too high hydraulic conductivity on our results, we repeated our simulations with a new value for the sheet conductivity parameter, $k_s = 1.0e^{-4}$ m$^{7/4}$ kg$^{-1/2}$ (for more details, see response to next comment below).

In contrast to our previous results, in our new simulations (Fig. S1) mean water pressures are higher (above 50% of overburden pressure, compared to near 0% in the previous simulations) throughout winter, suggesting that the subglacial drainage system remains too inefficient to entirely drain the meltwater before the start of the next melt season. However, the wintertime subglacial drainage system is not sufficiently pressurized and its discharge is too low to sustain year-round channels.

**Specific comments**

**(3) How much of an influence do your parameter choices (hydraulic conductivities) have on the qualitative behaviour of your results?**

*Response:* Our value for the sheet conductivity (ks) is lower than the one originally proposed by Werder et al. (2013), but inspection of current literature revealed that others find better agreement to observations by using lower values (e.g. Dow et al., 2020) or even suggest a seasonally variable conductivity (Downs et al., 2018). Werder et al. (2013) simulated a single melt season and hence they did not experience the sheet running dry over winter (although they discuss this shortcoming), whereas multi-year simulations by the other studies explicitly address this problem. We agree with the reviewer that too high values for the sheet conductivity (ks) may thus lead to unrealistic results, mainly with regard to the basal water pressure that can also impact the degree of channelization of the subglacial hydrological system. However, we would like to emphasize that the aim of our study is to investigate the effect of different input configurations during the melt season rather than to obtain most realistic winter pressures.

Nevertheless, to test whether our results would be affected by adopting different values for ks and kc, we conducted sensitivity tests over the first three years of our 15-year long simulations. Whereas the overall results are largely insensitive to the choice of kc within reasonable limits (ks/kc ratio $> 1.0e^{-2}$, with ks = $[1.0e^{-3}; 1.0e^{-4}]$ m$^{7/4}$ kg$^{-1/2}$ and kc = $[1.0e^{-1}; 1.0e^{-2}]$ m$^{3/2}$ kg$^{-1/2}$), lowering ks results in higher mean water pressures (above 50% of overburden pressure) throughout winter (Fig. S1), as well as in a more developed channel network (longer channels and higher connectivity) during the melt season. However, lowering ks also leads to substantially higher mean water pressures (above 60% of overburden pressure) during the entire melt season (Fig. S1), suggesting that the channelized drainage system indeed lacks efficiency and can only exist at high water pressure. While this increases the realism of our simulations, we also find that our original conclusions about limited influence of channelization and anti-clockwise pressure-input—hysteresis are robust, and hence are also the conclusions about the role of different recharge configurations.

We are now running the full 15-year simulations for all four experiments with a sheet conductivity (ks) of $1.0e^{-4}$ m$^{7/4}$ kg$^{-1/2}$, which yields more realistic winter water pressures, and we will update our results with these model outputs.

**(4) A final question that, again I just wonder if you have any thoughts on, is what is the missing ingredient that would produce more efficient drainage systems at the Kongsfjord basin? Is it simply that the meltwater supplied to the subglacial hydrology system is too small, or that the parameter values are insufficient. If, for example, you could choose conductivity values that would give "efficient drainage", are they too far away from the physical values to be plausible?**

*Response:* Our sensitivity tests revealed that changing the value of the sheet and channel conductivities has limited effects on the efficiency of the channelized drainage system. Glacier geometry (gentle slopes and relatively thick ice) and the short duration and low intensity of meltwater production are more likely the main limiting factors of channel efficiency. It might be interesting to further investigate these effects in order to quantify them, however this is outside the scope of this study.

**Technical corrections**

**Line 69) Colon misuse, I think you can just remove it.**

*Response:* Colon was removed.

**Line 70) I think "approximations on" should be something more like "approximations about" or "approximations of".**

*Response:* "Approximations on" was changed to "approximations about".

**Line 98) Should be a double hyphen (en dash) between "balance" and "snow".**

*Response:* Fixed.

**Line 151) Colon misuse (clause before colon should be independent). I also think semicolons should be used to separate list items here because there is internal grammar in each of the items (commas).**

*Response:* Colon was removed.

**Line 323) "channels do no align"-> "channels do not align".**

*Response:* Fixed.

**Line 434 + 502) Capitalisation of "arctic" (-> "Arctic").**

*Response:* Fixed.

**Line 434) "disadvantages chanelization" seems a bit awkward, maybe "inhibits chanelization" or "prevents chanelization" (I think this one is just personal taste so happy for it to be ignored).**

*Response:* "disadvantages channelization" was changed to "inhibits channelization".

**References**

Dow, C. F., McCormack, F. S., Young, D. A., Greenbaum, J. S., Roberts, J. L., & Blankenship, D. D. (2020). Totten Glacier subglacial hydrology determined from geophysics and modeling. *Earth and Planetary Science Letters*, *531*, 115961.

Downs, J. Z., Johnson, J. V., Harper, J. T., Meierbachtol, T., & Werder, M. A. (2018). Dynamic hydraulic conductivity reconciles mismatch between modeled and observed winter subglacial water pressure. *Journal of Geophysical Research: Earth Surface*, *123*(4), 818-836.

Koziol, C. P., & Arnold, N. (2018). Modelling seasonal meltwater forcing of the velocity of land-terminating margins of the Greenland Ice Sheet. *The Cryosphere*, *12*(3), 971-991.

Werder, M. A., Hewitt, I. J., Schoof, C. G., & Flowers, G. E. (2013). Modeling channelized and distributed subglacial drainage in two dimensions. *Journal of Geophysical Research: Earth Surface*, *118*(4), 2140-2158.

[Figure]

**Figure S1.** Mean (2004–2005) annual **(a)** water input and **(b)** basal water pressure averaged over the whole model domain for each experiment. The shaded area is the standard deviation showing the interannual variability of water input and water pressure for each experiment. Based on Figure 3 from the manuscript.

---

## Author Comment (AC2) · 23 Feb 2021

Main points

**(1) I do, unfortunately, have a major concern, which is that the subglacial water pressure is far too low. It drops down to almost 0% of overburden during winter which is very unrealistic and then in the summer the mean pressure is less than 80%. From Figure 8, it looks like only a very small region of your domain gets to overburden pressure with the rest significantly lower. As a reference, boreholes that hit efficient systems often have pressure varying 20-60% overburden and that is considered low pressure. The distributed system should have high pressure, which would be anything above about 80% overburden. The reason that you have the seasonal and channelization behaviour that you see is that the model is spending most of the season building up to a background level of pressurization, which you would normally assume it would already have at the beginning of the season. Instead, with the spring event, water inputs should be into a system already close to overburden pressure. The rapid ice acceleration during this time is because often the basal system will increase to pressures above overburden, hydraulically jacking up the ice and allowing fast flow. I notice that you don't note in the manuscript how you spin up the model. For GlaDS (and many other models), you have to have a spin-up period so that the system can adjust to the background inputs, which in this case should be whatever basal water is available. You then have to make sure your chosen parameters allow as realistic a system as possible for when you initiate you seasonal inputs. This is why sensitivity tests are often used to assess the variations that parameters will have on the system and, with GlaDS, the two most important parameters to test are the sheet and channel conductivities. Set either too high and the system won't pressurize and is unrealistic. Set either too low and the model will break because the system will become too pressurized. From those sensitivity tests you then have a range of applicable conductivity values to use as a starting off point for your four experiments. If you have a look at the GlaDS literature for the Antarctic (Dow et al, 2018; Dow et al, 2020) and from Greenland (Poinar et al, 2019; Cook et al, 2020), you'll see that the sheet and channels conductivity values used are much lower (<1e-4 and <0.1, respectively) than have been applied in your study, which explains the low pressures throughout your domain.**

*Response:* The reviewer raised concerns about unrealistically low water pressures before the onset of melting in our simulations, and further suggested that this behaviour was controlled by a too high hydraulic conductivity attributed to the sheet in our simulations.

Our value for the sheet conductivity ($k_s$) is lower than the one originally proposed by Werder et al. (2013), but inspection of current literature revealed that others find better agreement to observations by using lower values (e.g. Dow et al., 2020) or even suggest a seasonally variable conductivity (Downs et al., 2018). Werder et al. (2013) simulated a single melt season and hence they did not experience the sheet running

dry over winter (although they discuss this shortcoming), whereas multi-year simulations by the other studies explicitly address this problem. We agree with the reviewer that either too high or too low values for the sheet conductivity (ks) may thus lead to unrealistic results. However, we would like to emphasize that the aim of our study is to investigate the effect of different input configurations during the melt season rather than to obtain most realistic winter pressures.

Nevertheless, to test whether our results would be affected by adopting different values for ks and kc, we conducted sensitivity tests over the first three years of our 15-year long simulations. Whereas the overall results are largely insensitive to the choice of kc within reasonable limits (ks/kc ratio $> 1.0e^{-2}$, with ks = $[1.0e^{-3}; 1.0e^{-4}]$ m$^{7/4}$ kg$^{-1/2}$ and kc = $[1.0e^{-1}; 1.0e^{-2}]$ m$^{3/2}$ kg$^{-1/2}$), lowering ks results in higher mean water pressures (above 50% of overburden pressure) throughout winter (Fig. S1), as well as in a more developed channel network (longer channels and higher connectivity) during the melt season. However, lowering ks also leads to substantially higher mean water pressures (above 60% of overburden pressure) during the entire melt season (Fig. S1), suggesting that the channelized drainage system indeed lacks efficiency and can only exist at high water pressure. While this increases the realism of our simulations, we also find that our original conclusions about limited influence of channelization and anti-clockwise pressure-input—hysteresis are robust, and hence are also the conclusions about the role of different recharge configurations.

We are now running the full 15-year simulations for all four experiments with a sheet conductivity (ks) of $1.0e^{-4}$ m$^{7/4}$ kg$^{-1/2}$, which yields more realistic winter water pressures, and we will update our results with these model outputs.

Regarding the spin up period, in our original simulations presented in the manuscript, the spin up period was very short, that is, after only a few days into the simulation water pressures were similar to water pressures of the following winter seasons (close to 0% of overburden pressure). This is why we included the spin up period in our analysis. However, in our new simulations with ks = $1.0e^{-4}$ m$^{7/4}$ kg$^{-1/2}$, the spin up period is longer as the wintertime water pressures of the first year are significantly lower than those of the next winter seasons. Therefore, in the revised analysis of our results, we will disregard the first year of the 15 simulation years in order to have a one-year spin up period.

**(2) GlaDS also tends to have some issues with over-winter pressures if a spatially and temporally uniform basal sliding speed is used. This is because the basal sliding is applied in the cavity opening term. I would recommend taking the basal sliding rate as a percentage of the surface sliding rate to get the spatial variability, and then adjust this temporally using records of summer vs. winter velocity (if you have them). The latter doesn't have to be high temporal resolution but a lower sliding speed in winter causing less cavity opening will allow the system to repressurise (that's assuming that winter sliding speeds are lower than in the spring).**

*Response:* Our sensitivity tests showed that adopting a lower value for the sheet conductivity (ks) allowed higher water pressures in the winter, so we consider this issue fixed. Again, we would like to emphasize that the focus of this study is on the effects of meltwater input configurations rather than on reproducing subglacial hydrological conditions outside the melt season.

Moreover, the glaciers of the Kongsfjord basin are, to some extent, likely soft-bedded and thus it is unlikely that feedback mechanisms between basal sliding and cavity opening dominate the overall drainage system at these glaciers, since cavities require hard bed conditions to open. The existence of a feedback mechanism between sliding speed and drainage efficiency for this kind of glaciers may not be adequately represented in GlaDS. This is why we chose to avoid introducing any additional complexity that may not be relevant in our case.

**(3) My final main point is that, on the assumption you have access to ice surface velocities for the region, that is the best way to test whether the model is correctly representing your study region. Even a spatially averaged mean velocity should generally match the mean water pressure records that you show in Figure 3b. If these have the same pattern it would make your arguments about the subglacial system evolution stronger.**

*Response:* In the revised version of the manuscript, we now qualitatively discuss how the surface velocity fields estimated in Schellenberger et al. (2015) compare with our modelled water pressure. While the fast-flowing outlets Kongsbreen North and Kronebreen coincide with modelled high water pressures, high water pressures in upstream regions (towards Isachsenfonna and Holtedahlfonna) do not correspond to higher surface velocities. This does not necessarily contradict our results since these regions are flat with very low driving stress that would lead to low surface velocity even for higher water pressures. Instead, this emphasizes that direct comparison between surface velocity and water pressure is not always relevant since ice flow also depends on glacier geometry. Validation of our results using velocity measurements would thus require an ice flow model, which is beyond the scope of our study.

**Specific comments**

**Line 61) Why would the hydraulic potential minimum seed channels normally?**

*Response:* By this sentence, we meant that simple theories for determining channel flow path are based on estimating the minimum hydraulic potential pathway. When surface meltwater is provided uniformly to the bed, channels will preferentially form along this pathway concentrating the water discharge, whereas local input from moulins is able to create channels anywhere by arbitrarily concentrating water flux. We clarified this part in the revised manuscript.

**Line 157) "which participated". Also what do you mean by distributed model in this sentence?**

*Response:* Fixed. Also "distributed" was changed to "two-dimensional".

**Line 178) When you say HP 'set to zero' how do you apply that? As tidewater glaciers the outlet boundary condition would be best set at overburden but it's not clear if you do this.**

*Response:* We actually do set the hydraulic potential at overburden. This boundary condition allows imposing water pressure to be equal to sea pressure at the depth of the outlet. Indeed, at the outlets we have

$$\varphi = \varphi_m + p_w = \rho_w g z_b + \rho_w g (z_{sl} - z_b) = 0$$

where $\varphi$ is the hydraulic potential, $\varphi_m$ the elevation potential, $p_w$ the sea water pressure, $z_b$ the bed elevation, and $z_{sl}$ the sea level elevation ($= 0$ m a.s.l.).

**Lines 197 and 373) Need some references for this statement. Most recent subglacial hydrology studies do use moulin inputs.**

*Response:* "This approximation is most commonly made" was changed to "This approximation is still commonly made". References were added (e.g. Cook et al., 2020).

**Line 206) Why only keep 10? How does this turn into 13?**

*Response:* That sentence was indeed a bit confusing, therefore we changed it to "The other eight moulins were manually detected on high-resolution aerial images derived from TopoSvalbard (https://toposvalbard.npolar.no/, Norwegian Polar Institute)."

**Line 265) It would be useful to know what that input is in m3/s in addition to the cumulative input for the total catchment that you state.**

*Response:* We added this value in the revised manuscript.

**Line 270) Why are the moulins only higher up (I may have missed this)?**

*Response:* In contrast to Experiment 2, in Experiment 3 moulins only receive meltwater that is produced in their upstream watersheds. Therefore, the meltwater that is produced downstream of moulins is not taken into account in Experiment 3. Except for Kongsvegen, we did not detect any moulins in the lower parts of the other Kongsfjorden glaciers as these areas are highly crevassed. Meltwater input through crevasses is only accounted for in Experiment 4. We clarified this in the revised manuscript.

**Line 289) State what output result it supports rather than what figure.**

*Response:* "This figure supports Fig. 3(b) and Fig. 4" was changed to "This figure supports model results for basal water pressure (Fig. 3(b) and Fig. 4)".

**Line 314) What kind of numerical artefacts? Why would these occur?**

*Response:* Short channel segments sometimes display instable behaviour and grow unrealistically large. Since GlaDS is built under the assumption of water-saturated channels, these local instabilities produce locally high discharge due to unrealistic channel radius. We clarified this in the revised manuscript.

**Data availability – model outputs not provided.**

*Response:* The underlying model outputs for all the figures presented in this paper will be deposited in a common public data repository after the manuscript has been revised.

**Figure 1 – more detail needed for your below-sea-level elevations in panel b).**

*Response:* Panel (b) was changed to add more detail to the below-sea level elevations, and colour of the line annotations was changed to improve the readability of the map.

**Figure 2 – you have a lot of moulins on boundary points. That might cause problems if you reduce the conductivity to get the system closer to overburden.**

*Response:* We did not encounter any problems in our sensitivity test runs.

**Table 1 – basal sliding speed would be better stated in m a-1 and match what you say in the text.**

*Response:* We kept the basal sliding speed stated in $m\ s^{-1}$ in the table to keep consistency with the input to the model, but added the value in $m\ s^{-1}$ in the text.

**References**

*Response:* The references mentioned by the reviewer will be added to the manuscript.

Cook, S. J., Christoffersen, P., Todd, J., Slater, D., & Chauché, N. (2020). Coupled modelling of subglacial hydrology and calving-front melting at Store Glacier, West Greenland. *The Cryosphere*, *14*(3), 905-924.

Dow, C. F., McCormack, F. S., Young, D. A., Greenbaum, J. S., Roberts, J. L., & Blankenship, D. D. (2020). Totten Glacier subglacial hydrology determined from geophysics and modeling. *Earth and Planetary Science Letters*, *531*, 115961.

Downs, J. Z., Johnson, J. V., Harper, J. T., Meierbachtol, T., & Werder, M. A. (2018). Dynamic hydraulic conductivity reconciles mismatch between modeled and observed winter subglacial water pressure. *Journal of Geophysical Research: Earth Surface*, *123*(4), 818-836.

Schellenberger, T., Dunse, T., Kääb, A., Kohler, J., & Reijmer, C. H. (2015). Surface speed and frontal ablation of Kronebreen and Kongsbreen, NW Svalbard, from SAR offset tracking. *The Cryosphere*, *9*(6), 2339-2355.

Werder, M. A., Hewitt, I. J., Schoof, C. G., & Flowers, G. E. (2013). Modeling channelized and distributed subglacial drainage in two dimensions. *Journal of Geophysical Research: Earth Surface*, *118*(4), 2140-2158.

[Figure]

**Figure S1.** Mean (2004–2005) annual **(a)** water input and **(b)** basal water pressure averaged over the whole model domain for each experiment. The shaded area is the standard deviation showing the interannual variability of water input and water pressure for each experiment. Based on Figure 3 from the manuscript.

---

## Author Response (AR1)

*Authors' response and revisions on* **"Sensitivity of subglacial drainage to water supply distribution at the Kongsfjord basin, Svalbard"**

*Authors:* Chloé Scholzen, Thomas V. Schuler, Adrien Gilbert

**Authors' changes to the manuscript**

Following the reviewers' comments, we made substantial changes to our manuscript. We re-ran all of our simulations with a new value for the sheet conductivity parameter (ks), and we removed the first year from our analysis in order to disregard the spin-up period. We remade all of the figures that contain model results (Fig. 3–9) and rewrote all the sections that include descriptions of the new results (Sect. 4.1, 4.2, 5.1, 5.2, 5.3, as well as the abstract).

We found that our conclusions regarding the sensitivity of the modelled subglacial hydrology to water input distribution and the efficiency of the subglacial drainage system at Kongsfjord remain valid with our new model results. Therefore, we made only minor edits to the discussion and conclusion sections of the manuscript. These changes include specific clarifications requested by the reviewers, as well as changes in the wording and in the structure so that the text reads more easily.

We addressed all specific comments by the reviewers and updated the manuscript accordingly.

**Authors' response to Anonymous Referee #1**

We thank the reviewer for their thorough and helpful review. We addressed their comments (shown in bold) point by point. Please also note the figure at the end of the document.

**General comments**

**(1) It seems like some of the findings are in line with those of Koziol and Arnold 2018 (modelling seasonal meltwater forcing on the velocity of land-terminating margins of the Greenland ice-sheet). In this paper they consider how a subglacial drainage system might contribute to decadal-timescale slowdown of ice in the ablation zone. One theory is that if a more efficient drainage system develops in the summer, the effective pressure throughout the winter will be higher (i.e. the water pressure will be lower), leading to slow down. Your results (figure 3b) seem to suggest that supraglacial-meltwater configurations that produce more efficient drainage (in summer) lead to lower water pressures in winter (as Koziol and Arnold suggest).**

*Response:* We thank the reviewer for mentioning this paper. Our results are indeed in accordance with those of Koziol and Arnold (2018) in that the configurations that produce more efficient drainage in summer lead to lower water pressures in winter. This is mainly visible in comparing Experiment 1 and Experiment 4, as the latter yields lower water pressures than the former during both summer and winter by developing a more efficient drainage in the upper half of the model domain. This paper has been appropriately referenced in the revised manuscript to highlight this point (lines 421–422).

**(2) You conclude that the resulting subglacial drainage systems that evolve are inefficient. I think here efficiency refers to how the water pressure responds to meltwater input. It seems though (I may**

**be wrong) that the drainage systems are efficient enough to entirely drain the meltwater before the next melt season. Thus, we have the "worst of both worlds" in that the drainage system is inefficient enough for a spike in water pressure at the start of every melt season, but efficient enough to completely drain and revert to it's original (most inefficient) state before the start of the next melt season.**

*Response:* Following the comments of the reviewers on the possible impact of a too high hydraulic conductivity on our results, we repeated our simulations with a new value for the sheet conductivity parameter, $k_s = 1.0e^{-4}$ $m^{7/4}$ $kg^{-1/2}$ (for more details, see response to next comment below).

In contrast to our previous results, in our new simulations mean water pressures are higher (above 50% of overburden pressure, compared to near 0% in the previous simulations) throughout winter, suggesting that the subglacial drainage system remains too inefficient to entirely drain the meltwater before the start of the next melt season. However, the wintertime subglacial drainage system is not sufficiently pressurized and its discharge is too low to sustain year-round channels.

**Specific comments**

**(3) How much of an influence do your parameter choices (hydraulic conductivities) have on the qualitative behaviour of your results?**

*Response:* Our value for the sheet conductivity ($k_s$) is lower than the one originally proposed by Werder et al. (2013), but inspection of current literature revealed that others find better agreement to observations by using lower values (e.g. Dow et al., 2018; Dow et al., 2020) or even suggest a seasonally variable conductivity (Downs et al., 2018). Werder et al. (2013) simulated a single melt season and hence they did not experience the sheet running dry over winter (although they discuss this shortcoming), whereas multi-year simulations by the other studies explicitly address this problem. We agree with the reviewer that too high values for the sheet conductivity ($k_s$) may thus lead to unrealistic results, mainly with regard to the basal water pressure that can also impact the degree of channelization of the subglacial hydrological system. However, we would like to emphasize that the aim of our study is to investigate the effect of different input configurations during the melt season rather than to obtain most realistic winter pressures.

Nevertheless, to test whether our results would be affected by adopting different values for $k_s$ and $k_c$, we conducted sensitivity tests over the first three years of our 15-year long simulations. Whereas the overall results are largely insensitive to the choice of $k_c$ within reasonable limits ($k_s/k_c$ ratio $> 1.0e^{-2}$, with $k_s = [1.0e^{-3}; 1.0e^{-4}]$ $m^{7/4}$ $kg^{-1/2}$ and $k_c = [1.0e^{-1}; 1.0e^{-2}]$ $m^{3/2}$ $kg^{-1/2}$), lowering $k_s$ results in higher mean water pressures (above 50% of overburden pressure) throughout winter, as well as in a more developed channel network (longer channels and higher connectivity) during the melt season. However, lowering $k_s$ also leads to substantially higher mean water pressures (above 60% of overburden pressure) during the entire melt season, suggesting that the channelized drainage system indeed lacks efficiency and can only exist at high water pressure. While this increases the realism of our simulations, we also find that our original conclusions about limited influence of channelization and anti-clockwise pressure-input—hysteresis are robust, and hence are also the conclusions about the role of different recharge configurations.

We have re-run our 15-year simulations for all four experiments with a sheet conductivity ($k_s$) of $1.0e^{-4}$ $m^{7/4}$ $kg^{-1/2}$ and we have updated our results with these new model outputs.

**(4) A final question that, again I just wonder if you have any thoughts on, is what is the missing ingredient that would produce more efficient drainage systems at the Kongsfjord basin? Is it simply**

that the meltwater supplied to the subglacial hydrology system is too small, or that the parameter values are insufficient. If, for example, you could choose conductivity values that would give "efficient drainage", are they too far away from the physical values to be plausible?

*Response:* Our sensitivity tests revealed that changing the value of the sheet and channel conductivities has limited effects on the efficiency of the channelized drainage system. Glacier geometry (gentle slopes and relatively thick ice) and the short duration and low intensity of meltwater production are more likely the main limiting factors of channel efficiency. It might be interesting to further investigate these effects in order to quantify them, however this is outside the scope of this study.

**Technical corrections**

**Line 69) Colon misuse, I think you can just remove it.**

*Response:* Colon was removed.

**Line 70) I think "approximations on" should be something more like "approximations about" or "approximations of".**

*Response:* "Approximations on" was changed to "approximations about".

**Line 98) Should be a double hyphen (en dash) between "balance" and "snow".**

*Response:* Fixed.

**Line 151) Colon misuse (clause before colon should be independent). I also think semicolons should be used to separate list items here because there is internal grammar in each of the items (commas).**

*Response:* Colon was removed.

**Line 323) "channels do no align"-> "channels do not align".**

*Response:* Fixed.

**Line 434 + 502) Capitalisation of "arctic" (-> "Arctic").**

*Response:* Fixed.

**Line 434) "disadvantages chanelization" seems a bit awkward, maybe "inhibits chanelization" or "prevents chanelization" (I think this one is just personal taste so happy for it to be ignored).**

*Response:* "disadvantages channelization" was changed to "inhibits channelization".

**Authors' response to Christine Dow, Referee #2**

We thank Christine Dow for her thorough and helpful review. We addressed her comments (shown in bold) point by point. Please also note the figure at the end of the document.

**Main points**

**(1) I do, unfortunately, have a major concern, which is that the subglacial water pressure is far too low. It drops down to almost 0% of overburden during winter which is very unrealistic and then in the summer the mean pressure is less than 80%. From Figure 8, it looks like only a very small region of your domain gets to overburden pressure with the rest significantly lower. As a reference, boreholes that hit efficient systems often have pressure varying 20-60% overburden and that is considered low pressure. The distributed system should have high pressure, which would be anything above about 80% overburden. The reason that you have the seasonal and channelization behaviour that you see is that the model is spending most of the season building up to a background level of pressurization, which you would normally assume it would already have at the beginning of the season. Instead, with the spring event, water inputs should be into a system already close to overburden pressure. The rapid ice acceleration during this time is because often the basal system will increase to pressures above overburden, hydraulically jacking up the ice and allowing fast flow. I notice that you don't note in the manuscript how you spin up the model. For GlaDS (and many other models), you have to have a spin-up period so that the system can adjust to the background inputs, which in this case should be whatever basal water is available. You then have to make sure your chosen parameters allow as realistic a system as possible for when you initiate you seasonal inputs. This is why sensitivity tests are often used to assess the variations that parameters will have on the system and, with GlaDS, the two most important parameters to test are the sheet and channel conductivities. Set either too high and the system won't pressurize and is unrealistic. Set either too low and the model will break because the system will become too pressurized. From those sensitivity tests you then have a range of applicable conductivity values to use as a starting off point for your four experiments. If you have a look at the GlaDS literature for the Antarctic (Dow et al, 2018; Dow et al, 2020) and from Greenland (Poinar et al, 2019; Cook et al, 2020), you'll see that the sheet and channels conductivity values used are much lower (<1e-4 and <0.1, respectively) than have been applied in your study, which explains the low pressures throughout your domain.**

*Response:* The reviewer raised concerns about unrealistically low water pressures before the onset of melting in our simulations, and further suggested that this behaviour was controlled by a too high hydraulic conductivity attributed to the sheet in our simulations.

Our value for the sheet conductivity (ks) is lower than the one originally proposed by Werder et al. (2013), but inspection of current literature revealed that others find better agreement to observations by using lower values (e.g. Dow et al., 2018; Dow et al., 2020) or even suggest a seasonally variable conductivity (Downs et al., 2018). Werder et al. (2013) simulated a single melt season and hence they did not experience the sheet running dry over winter (although they discuss this shortcoming), whereas multi-year simulations by the other studies explicitly address this problem. We agree with the reviewer that either too high or too low values for the sheet conductivity (ks) may thus lead to unrealistic results. However, we would like to emphasize that the aim of our study is to investigate the effect of different input configurations during the melt season rather than to obtain most realistic winter pressures.

Nevertheless, to test whether our results would be affected by adopting different values for ks and kc, we conducted sensitivity tests over the first three years of our 15-year long simulations. Whereas the overall

results are largely insensitive to the choice of kc within reasonable limits (ks/kc ratio > $1.0e^{-2}$, with ks = $[1.0e^{-3}; 1.0e^{-4}]$ $m^{7/4}$ $kg^{-1/2}$ and kc = $[1.0e^{-1}; 1.0e^{-2}]$ $m^{3/2}$ $kg^{-1/2}$), lowering ks results in higher mean water pressures (above 50% of overburden pressure) throughout winter, as well as in a more developed channel network (longer channels and higher connectivity) during the melt season. However, lowering ks also leads to substantially higher mean water pressures (above 60% of overburden pressure) during the entire melt season, suggesting that the channelized drainage system indeed lacks efficiency and can only exist at high water pressure. While this increases the realism of our simulations, we also find that our original conclusions about limited influence of channelization and anti-clockwise pressure-input—hysteresis are robust, and hence are also the conclusions about the role of different recharge configurations.

We have re-run our 15-year simulations for all four experiments with a sheet conductivity (ks) of $1.0e^{-4}$ $m^{7/4}$ $kg^{-1/2}$ and we have updated our results with these new model outputs.

Regarding the spin-up period, in our original simulations presented in the manuscript, the spin-up period was very short, that is, after only a few days into the simulation water pressures were similar to water pressures of the following winter seasons (close to 0% of overburden pressure). This is why we included the spin-up period in our analysis. However, in our new simulations with ks = $1.0e^{-4}$ $m^{7/4}$ $kg^{-1/2}$, the spin-up period is longer as the wintertime water pressures of the first year are significantly lower than those of the next winter seasons. Therefore, in the revised analysis of our results, we have disregarded the first year of the 15 simulation years in order to have a one-year spin-up period (line 260).

**(2) GlaDS also tends to have some issues with over-winter pressures if a spatially and temporally uniform basal sliding speed is used. This is because the basal sliding is applied in the cavity opening term. I would recommend taking the basal sliding rate as a percentage of the surface sliding rate to get the spatial variability, and then adjust this temporally using records of summer vs. winter velocity (if you have them). The latter doesn't have to be high temporal resolution but a lower sliding speed in winter causing less cavity opening will allow the system to repressurise (that's assuming that winter sliding speeds are lower than in the spring).**

*Response:* Our sensitivity tests showed that adopting a lower value for the sheet conductivity (ks) allowed higher water pressures in the winter, so we consider this issue fixed. Again, we would like to emphasize that the focus of this study is on the effects of meltwater input configurations rather than on reproducing subglacial hydrological conditions outside the melt season.

Moreover, the glaciers of the Kongsfjord basin are, to some extent, likely soft-bedded and thus it is unlikely that feedback mechanisms between basal sliding and cavity opening dominate the overall drainage system at these glaciers, since cavities require hard bed conditions to open. The existence of a feedback mechanism between sliding speed and drainage efficiency for this kind of glaciers may not be adequately represented in GlaDS. This is why we chose to avoid introducing any additional complexity that may not be relevant in our case.

**(3) My final main point is that, on the assumption you have access to ice surface velocities for the region, that is the best way to test whether the model is correctly representing your study region. Even a spatially averaged mean velocity should generally match the mean water pressure records that you show in Figure 3b. If these have the same pattern it would make your arguments about the subglacial system evolution stronger.**

*Response:* In the revised version of the manuscript, we now qualitatively discuss how the surface velocity fields estimated by Schellenberger et al. (2015) compare with our modelled water pressure (lines 477–482). The fast-flowing outlet Kronebreen exhibits higher surface velocity during summer, which coincides with our modelled high water pressures during the entire melt season. However, Schellenberger et al. (2015) show that the Kongsbreen North outlet behaves differently, with a sharp drop in summertime surface velocity that is not captured by our model in terms of water pressure. We interpret this discrepancy as an increase in drainage efficiency due to permanent canals carved in the sediment, a mechanism that is not accounted for by the Röthlisberger theory of channels formation as implemented in GlaDS.

**Specific comments**

**Line 61) Why would the hydraulic potential minimum seed channels normally?**

*Response:* By this sentence, we meant that simple theories for determining channel flow path are based on estimating the minimum hydraulic potential pathway. When surface meltwater is provided uniformly to the bed, channels will preferentially form along this pathway concentrating the water discharge, whereas local input from moulins is able to create channels anywhere by arbitrarily concentrating water flux. We clarified this part in the revised manuscript (lines 60–61).

**Line 157) "which participated". Also what do you mean by distributed model in this sentence?**

*Response:* Fixed. Also "distributed" was changed to "two-dimensional".

**Line 178) When you say HP 'set to zero' how do you apply that? As tidewater glaciers the outlet boundary condition would be best set at overburden but it's not clear if you do this.**

*Response:* We actually do set the hydraulic potential at overburden. This boundary condition allows imposing water pressure to be equal to sea pressure at the depth of the outlet. Indeed, at the outlets we have

$$\varphi = \varphi_m + p_w = \rho_w g z_b + \rho_w g(z_{sl} - z_b) = 0$$

where $\varphi$ is the hydraulic potential, $\varphi_m$ the elevation potential, $p_w$ the sea water pressure, $z_b$ the bed elevation, and $z_{sl}$ the sea level elevation (= 0 m a.s.l.).

We added this equation in the revised manuscript (lines 178–182).

**Lines 197 and 373) Need some references for this statement. Most recent subglacial hydrology studies do use moulin inputs.**

*Response:* "This approximation is most commonly made" was changed to "This approximation is still commonly made". References were added (e.g. Cook et al., 2020).

**Line 206) Why only keep 10? How does this turn into 13?**

*Response:* That sentence was indeed a bit confusing, therefore we changed it to "The other eight moulins were manually detected on high-resolution aerial images derived from TopoSvalbard (https://toposvalbard.npolar.no/, Norwegian Polar Institute)."

**Line 265) It would be useful to know what that input is in m3/s in addition to the cumulative input for the total catchment that you state.**

*Response:* We changed the water flux values to total water volume values in the revised manuscript.

**Line 270) Why are the moulins only higher up (I may have missed this)?**

*Response:* In contrast to Experiment 2, in Experiment 3 moulins only receive meltwater that is produced in their upstream watersheds. Therefore, the meltwater that is produced downstream of moulins is not taken into account in Experiment 3. Except for Kongsvegen, we did not detect any moulins in the lower parts of the other Kongsfjorden glaciers as these areas are highly crevassed. Meltwater input through crevasses is only accounted for in Experiment 4. We clarified this in the revised manuscript (lines 217–219 and 222–223).

**Line 289) State what output result it supports rather than what figure.**

*Response:* "This figure supports Fig. 3(b) and Fig. 4" was changed to "This figure supports model results for basal water pressure (Fig. 3(b) and Fig. 4)".

**Line 314) What kind of numerical artefacts? Why would these occur?**

*Response:* Short channel segments sometimes display instable behaviour and grow unrealistically large. Since GlaDS is built under the assumption of water-saturated channels, these local numerical instabilities produce locally high discharge due to unrealistic channel radius. We clarified this in the revised manuscript (lines 327–328).

**Data availability – model outputs not provided.**

*Response:* The underlying model outputs for the figures presented in this paper can be accessed at https://doi.org/10.5281/zenodo.4680908.

**Figure 1 – more detail needed for your below-sea-level elevations in panel b).**

*Response:* Panel (b) was changed to add more detail to the below-sea level elevations, and colour of the line annotations was changed to improve the readability of the map.

**Figure 2 – you have a lot of moulins on boundary points. That might cause problems if you reduce the conductivity to get the system closer to overburden.**

*Response:* We did not encounter any problems in our sensitivity test runs.

**Table 1 – basal sliding speed would be better stated in m a-1 and match what you say in the text.**

*Response:* We kept the basal sliding speed stated in m s$^{-1}$ in the table to keep consistency with the input to the model, but added the value in m s$^{-1}$ in the text (line 191).

---

## Author Response (AR2)

*Authors' response and minor revisions on* **"Sensitivity of subglacial drainage to water supply distribution at the Kongsfjord basin, Svalbard"**

*Authors:* Chloé Scholzen, Thomas V. Schuler, Adrien Gilbert

**Authors' response to Referee #2**

We thank Christine Dow for her helpful suggestions (shown in bold), which we implemented in the new version of the manuscript. We believe that the manuscript has considerably gained in quality and in readability thanks to her thorough review.

**I would like to thank the authors for their adjustment of their manuscript and study in response to my review. Using the lower sheet conductivity makes the system look a lot more realistic and so I have much more faith in the outputs being produced. Again, including a spin-up year also helps with output validity. The results and analysis are well presented and are interesting for analysis of Svalbard Glaciers compared to lower latitude glaciers.**

**I have no major comments and my points below are primarily suggestions for tightening the language for clarity.**

**Language suggestions**

**9 – 'modulates ice velocity'.**

**16 – 'which instead is controlled…'**

**17 – 'which we attribute to small surface gradients'**

**18 – 'The findings of our study are potentially applicable to most Svalbard…'**

**31 – 'been found to be concurrent'**

**49 – Perhaps 'in reality' rather than 'In nature'. And 'usually is a result of water accumulation over a catchment on the glacier surface'.**

**53 – also by fractures (in addition to englacial conduits)**

**92 – 'nunataks are present in the lower parts' rather than 'peak through'**

**94 – 'ice thickens to a maximum of 740m'**

**109 – put 'measured in 2012' into the brackets with the reference as otherwise it suggests that the maximum speed was in 2012 and it has slowed since.**

**146 – 'drainage system was achieved'**

**152 – 'associated with'**

**155 – 'we apply the Glacier'**

**164** – what is the value of that uniform basal melt rate you apply?

**170** – 'we direct readers'

**184** – 'values different from Werder et al (2013)' Other literature discussing GlaDS uses these values.

**185** – 'below the channel'

**205** – suggest you replace 'supraglacial hydrology' with 'supraglacial drainage network'

**222** – 'associated with'

**237** – 'catchments are also adjusted in order to'

**242** – 'analysis was carried out'. Check rest of paragraph/manuscript for correct tenses.

**274** – 'different sizes of surface area contributing water'

**302** – 'this figure' still confusing. Delete 'supports model results for basal water pressure (Fig 3 and 4) an' so it reads 'This figures shows that, at…' Otherwise it is redundant as you're discussing the same model outputs.

**305** – 'typically approaches overburden pressure'

**308** – 'Over the annual period'

**309** – 'recurrent pattern in all seasons is'

**401** – 'However, results from this experiment'

**410** – 'less' instead of 'poorer'

**530** – 'less water supply to their beds compared to lower latitude glaciers'.

**533** – last sentence needs work. Instead of 'standard configuration' how about contrary to what would be expected in Greenland outlet or Alpine glaciers.

**Figure 7 caption 'subglacial channel distribution'**

*Response to all previous suggestions:* done.

**Other suggestions/questions**

**67 – 'distributed supply rates' is confusing**

*Response:* We removed the words 'distributed' (confusing) and 'calibrated' (not relevant), and added 'gridded' to convey that the input is spatially distributed.

**76 – '75% is currently ice-covered'**

*Response:* We replaced 'glacierised' with 'glacier-covered'.

**107 – this point about temperate ice is important as it raises the question of where it might not be temperate at the bed. It would be good to briefly expand on this, perhaps in the discussion.**

*Response:* We replaced the part "(...) which indicates that the glacier base is temperate" with "which is consistent with the finding of widespread temperate basal conditions at several glaciers in the region (Björnsson et al., 1996; Sevestre et al., 2015)".

**235 – not directly from Fig 2c. What were they specifically identified from? The supraglacial drainage network and…?**

*Response:* The sentence was changed to "These moulins were identified from the modelled supraglacial drainage network (Fig. 2(c)) and the observed crevasses (Fig. 2(d)); more specifically, we added a moulin wherever a supraglacial stream crosses the upper boundary of a crevassed area".

**346 – how does the Praminik hydrological analysis affect your confidence in your outputs vs the observations by How et al and Everett et al? Does the Praminik study back up or counter your GlaDS results and why?**

*Response:* The study by Pramanik et al. (2020) mainly focuses on the relative differences in subglacial discharge between Kongsbreen and Kronebreen; differences between Kronebreen North and Kronebreen South are barely mentioned. Therefore, we decided to remove this reference from our manuscript.

The study by How et al. (2017) focuses on Kronebreen only. Their observations indicate that the north-side plume is larger and temporally more stable than the south-side plume; however, they mention that the south-side plume was difficult to measure. Therefore, for the south-side outlet, comparing our model results to their observations remains inconclusive. For the north-side outlet, the observed stability of the plume indicates the existence of a persistent drainage pathway. This could presumably be due to a canal incised into the sediment (Walder and Fowler, 1994), a mechanism that is not included in our model, as mentioned in Sect. 5.3 and Sect. 5.4.

We changed that part to: "This is in disagreement with observations that suggest the north-side outlet has larger and more temporally stable plume activity than the south-side outlet (How et al., 2017; Everett et al., (2018)). While there are geometrical challenges with reliably measuring the south-side plumes area from time-lapse photography (How et al., 2017), the observed stability of the north-side plumes clearly indicates the existence of a persistent drainage pathway that is not captured by our model. This subglacial pathway could presumably exist in the form a permanent canal incised into the sediment (Walder and Fowler, 1994), a mechanism that is not included in GlaDS (as described in Sect. 5.4)".

**351 – I'm not sure how this is implied? Can you see evidence of that in your model results?**

*Response:* The sentence was indeed confusing, therefore we changed it to "In late August, channelized discharge in Experiments 1–3–4 reaches its peak, whereas in Experiment 2 the channel network is already collapsing in the regions that receive less water input".

In fact, upon closer inspection of our model results, we see that, in Experiment 2, the channelized drainage system systematically collapses sooner than in Experiments 1–3–4.

**384 – how rapid is the closure?**

*Response:* 'rapid' was changed to 'gradual'. The following sentence gives an approximate idea of how fast the channels close, by mentioning "at the glacier termini channels persist until late October".

**443 – 'channelization occurs only with high pressurisation'. I'm not sure that follows – you also have channelization when you applied your previous higher sheet conductivity with overall much lower pressure levels.**

*Response:* Even in the initial simulations (with higher sheet conductivity), channelization coincided with high water pressures. Indeed, although domain-wide averaged pressures were lower in the initial simulations (Fig. 3), local pressures were high during the melt season. This is clearly visible in Fig. 4(i) (version 1 of the manuscript), which shows water pressures between 70–100 % of ice overburden pressure in July in the ablation zone, and especially high in the regions where channels opened. Fig. 4(i) shows pressures for Experiment 1, which yields higher pressures overall; however, Fig. 4(j, k, l) show that, in the ablation zone, pressure biases between Experiments 2, 3, 4 and Experiment 1 are close to zero, indicating that in all four experiments water pressures are high in this part of the domain.